# Identification of a stereotypic molecular arrangement of endogenous glycine receptors at spinal cord synapses

**Stephanie A Maynard[1], Philippe Rostaing[1], Natascha Schaefer[2], Olivier Gemin[1], Adrien Candat[1], Andréa Dumoulin[1], Carmen Villmann[2], Antoine Triller[1]\*, Christian G Specht[1,3]\***

[1]Institut de Biologie de l'ENS (IBENS), École Normale Supérieure, CNRS, Inserm, PSL University, Paris, France; [2]Institute for Clinical Neurobiology, University Hospital, Julius-Maximilians-University, Wuerzburg, Germany; [3]Diseases and Hormones of the Nervous System (DHNS), Inserm U1195, Université Paris-Saclay, Paris, France

**Abstract** Precise quantitative information about the molecular architecture of synapses is essential to understanding the functional specificity and downstream signaling processes at specific populations of synapses. Glycine receptors (GlyRs) are the primary fast inhibitory neurotransmitter receptors in the spinal cord and brainstem. These inhibitory glycinergic networks crucially regulate motor and sensory processes. Thus far, the nanoscale organization of GlyRs underlying the different network specificities has not been defined. Here, we have quantitatively characterized the molecular arrangement and ultra-structure of glycinergic synapses in spinal cord tissue using quantitative super-resolution correlative light and electron microscopy. We show that endogenous GlyRs exhibit equal receptor-scaffold occupancy and constant packing densities of about 2000 GlyRs µm$^{-2}$ at synapses across the spinal cord and throughout adulthood, even though ventral horn synapses have twice the total copy numbers, larger postsynaptic domains, and more convoluted morphologies than dorsal horn synapses. We demonstrate that this stereotypic molecular arrangement is maintained at glycinergic synapses in the *oscillator* mouse model of the neuromotor disease hyperekplexia despite a decrease in synapse size, indicating that the molecular organization of GlyRs is preserved in this hypomorph. We thus conclude that the morphology and size of inhibitory postsynaptic specializations rather than differences in GlyR packing determine the postsynaptic strength of glycinergic neurotransmission in motor and sensory spinal cord networks.

**\*For correspondence:**
triller@biologie.ens.fr (AT);
christian.specht@inserm.fr (CGS)

**Competing interest:** The authors declare that no competing interests exist.

## Editor's evaluation

The manuscript presents a quantitative and comprehensive characterization of glycinergic synapses and spinal cord synapse organization that will be a highly valuable resource for the field. The use of cutting-edge imaging and quantitative techniques allows an understanding of these understudied synapses at a high level.

## Introduction

Synaptic transmission relies on the integration of spatially and temporally controlled signals by neurotransmitter receptors in the postsynaptic membrane. The molecular arrangement of postsynaptic receptors and scaffold proteins is therefore key to the synaptic function. However, the heterogeneity and complexity of postsynaptic sites have made it difficult to resolve its internal organization,

ascertain whether distinct compositional states exist, and determine how the organization is affected in disease.

Glycine receptors (GlyRs) are the main inhibitory neurotransmitter receptors in the adult spinal cord and brainstem. Glycinergic neurons arise from different embryonic origins, with specific types of neurons residing in characteristic layers of the spinal cord (*Lu et al., 2015*). Depending on their location, glycinergic neurons mediate sensory and motor information in the dorsal and ventral spinal cord, respectively, which requires high reliability and fidelity of transmission (*Alvarez, 2017*). Consequently, deficits in glycinergic transmission are involved in pain mechanisms (*Harvey et al., 2004*) and motor-related neurological diseases (*Schaefer et al., 2018*). The electrophysiological properties of glycinergic currents indicate that only a limited number of receptors are activated by the release of a single synaptic vesicle (*Oleskevich et al., 1999*; *Singer and Berger, 1999*), suggesting that the nanoscale organization of the receptors determines signal amplitude.

It has been shown that neurotransmitter receptors at excitatory and inhibitory synapses are organized within sub-synaptic domains (SSDs) that are aligned with presynaptic elements of the active zone (AZ) (*Crosby et al., 2019*; *MacGillavry et al., 2013*; *Pennacchietti et al., 2017*; *Tang et al., 2016*; *Yang et al., 2021*). These so-called trans-synaptic nanocolumns are thought to increase the efficacy of synaptic transmission (*Haas et al., 2018*). At mixed inhibitory synapses, both glycine and GABA$_A$ receptors are immobilized opposite to presynaptic release sites through direct interactions with their common scaffold protein gephyrin (*Maric et al., 2011*; *Specht et al., 2013*; *Yang et al., 2021*). However, accurate quantification of receptor numbers and their precise arrangement within postsynaptic sites in native tissue is lacking. Further, the question is raised as to whether the structure of glycinergic synapses varies in functionally diverse circuits of the dorsal and ventral spinal cord, whether it changes over time, and whether it is disturbed in GlyR pathologies such as the neuromotor disease hyperekplexia in humans. In hyperekplexic patients, mutations in the receptor subunit genes *GLRA1* and *GLRB* lead to decreased receptor availability and disturbances in glycinergic transmission, resulting in exaggerated startle reflexes, muscle hypertonia, and stiffness in infancy (*Chung et al., 2013*; *Chung et al., 2010*; *Schaefer et al., 2013*).

To investigate whether the molecular arrangement of GlyRs may account for differences in the functional specificity of sensory and motor circuits, we have quantitatively analyzed the ultra-structural organization of inhibitory synapses in spinal cord tissue. We have combined molecule counting of endogenous GlyRs using single-molecule localization microscopy (SMLM) with correlative light and electron microscopic analysis (CLEM) to obtain receptor numbers as well as detailed spatial information of the synapse at the nanometer scale. We have further examined to what extent the molecular organization is maintained throughout adult development and during GlyRα1 deficiency. We show that GlyRs are packed at a constant density of about 2000 receptor complexes per µm$^2$ at mature synapses, suggesting that they are assembled in a stereotypic fashion. This GlyR molecular organization is maintained in the hyperekplexia model *oscillator* despite a decrease in ventral synapse size, indicating that GlyRα1 deficiency does not affect the integrity of the synaptic arrangement as such. Collectively, our results provide the structural basis for understanding the mechanisms underlying receptor availability and the integration of neurotransmitter-induced signals.

## Results

### Generation of a KI model expressing endogenous levels of mEos4b-GlyRβ

In order to quantify GlyR numbers and their precise distribution at synapses, we generated a knock-in (KI) mouse model expressing endogenous mEos4b-tagged GlyRβ subunits (*Figure 1* and *Figure 1—figure supplement 1*). The β-subunit drives the synaptic localization of the receptor through direct interactions with the synaptic scaffold protein gephyrin at inhibitory synapses (*Kim et al., 2006*; *Meyer et al., 1995*). To date, labeling of GlyRβ in situ using immunocytochemistry has proven difficult due to a lack of reliable antibodies that recognize the native β-subunit (only antibodies for Western blotting recognizing the denatured protein are available), which has severely limited the study of the receptor. The coding sequence of mEos4b was inserted in exon 2 of the *Glrb* gene by homologous recombination supported by a guide RNA (CRISPR-Cas9) to increase recombination efficiency (ICS, Illkirch, France). Specifically, the fluorophore sequence was inserted after the signal peptide and before the

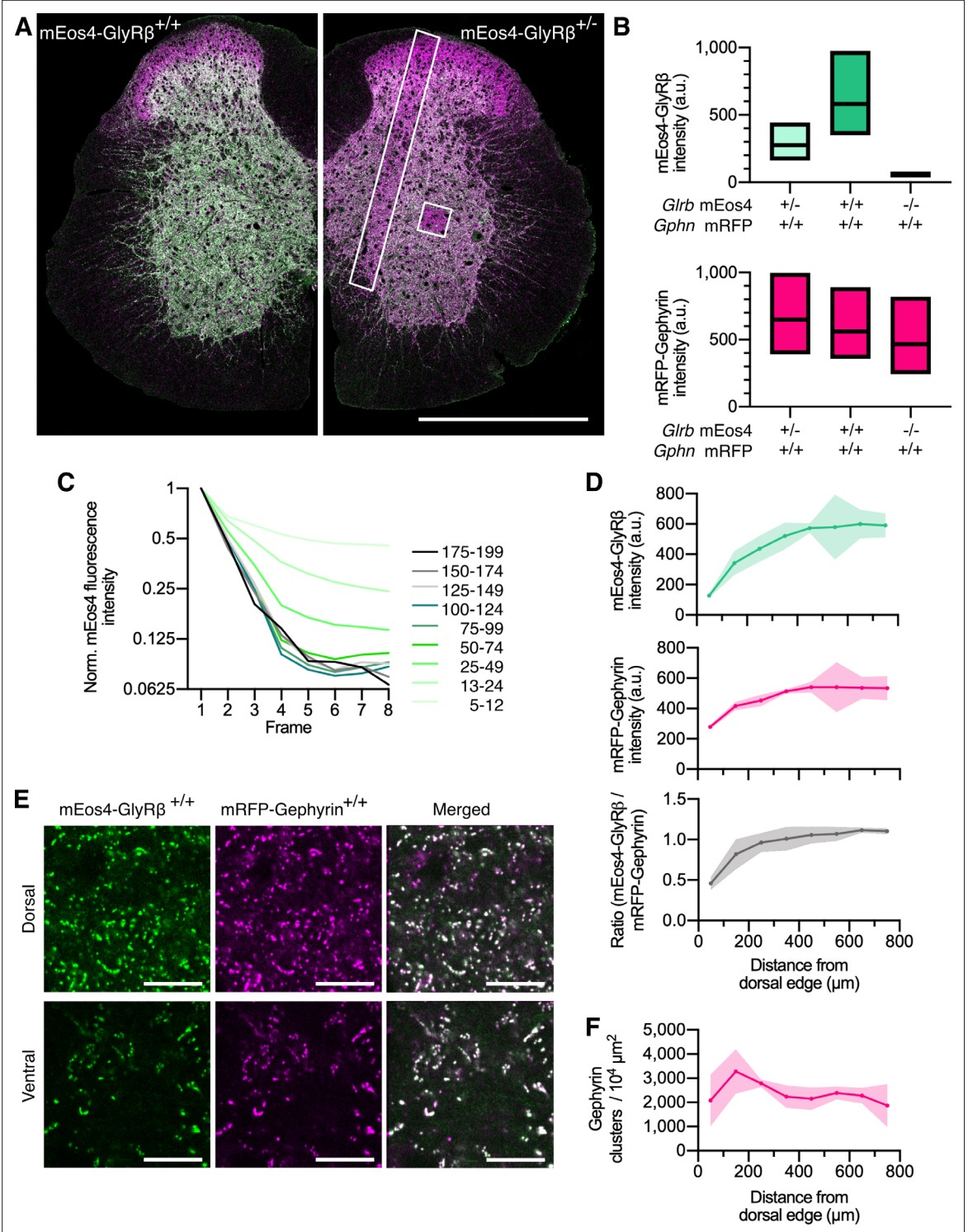

**Figure 1.** Quantitative confocal imaging of endogenous glycine receptors (GlyRs) in spinal cord tissue. (**A**) Representative confocal images of 40 µm spinal cord tissue sections from homozygous (+/+) and heterozygous (+/-) mEos4b-GlyRβ mice (green). Both mice are homozygous for mRFP-gephyrin (magenta). Scale bar = 0.5 mm. (**B**) Quantification of mEos4b-GlyRβ and mRFP-gephyrin fluorescent intensity of homozygous and heterozygous 2-month-old animals measured in the area indicated by the white square in (**A**). Plots show median and quartiles. N = 5–6 images from 5 to 6 tissue slices per genotype from 2 mice per age group. (**C**) Normalized fluorescent decay traces of homozygous mEos4b measured in the area indicated by the white square in (**A**) over eight consecutive frames. Intensities were binned in the first image and tracked on an individual synapse basis across the eight frames. (**D**) Mean intensity ± 95% confidence interval of mEos4b-GlyRβ and mRFP-gephyrin measured from distal edge of spinal cord in 2-month-old homozygous mice. Intensities measured in region as indicated by white rectangle in (**A**). N = 3 images from three tissue slices from two

*Figure 1 continued on next page*

*Figure 1 continued*

mice per condition. (**E**) Representative images of homozygous mEos4b-GlyRβ and mRFP-gephyrin at dorsal and ventral synapses. Scale bar = 10 μm. (**F**) Quantification of numbers of gephyrin clusters across the spinal cord. Plot shows mean ± 95% confidence interval. N = 3 images from three tissue slices from two mice per condition.

The online version of this article includes the following figure supplement(s) for figure 1:

**Figure supplement 1.** Generation of the mEos4b-GlyRβ knock-in mouse model.

**Figure supplement 2.** Physiological and functional characterization of mEos4b-GlyRβ knock-in mice.

**Figure supplement 3.** Protein expression of mEos4b-GlyRβ in brain sections of knock-in mice.

**Figure supplement 4.** Quantitative confocal imaging in 10 month old animals.

N-terminus of the mature GlyRβ subunit, meaning that it does not interrupt the coding sequence of the receptor (*Figure 1—figure supplement 1B*). The correct insertion was confirmed by amplification and sequencing of genomic DNA. Semi-quantitative RT-PCR revealed that equal concentrations of $Glrb^{Eos}$ and the wild-type (WT) transcript ($Glrb^{WT}$) are expressed in heterozygous animals. When bred to homozygosity, KI animals follow Mendelian inheritance (*Figure 1—figure supplement 2A*), exhibit normal lifespans (*Figure 1—figure supplement 2B*), and display no overt phenotype, suggesting that the GlyR expression and/or function are not altered.

To further confirm that GlyR function is not altered by the introduction of mEos4b, we carried out whole-cell recordings in cultured spinal cord neurons of $Glrb^{WT/WT}$ and homozygous $Glrb^{Eos/Eos}$ animals (*Figure 1—figure supplement 2C*). The agonist glycine was applied in a concentration series from 1 μM to 300 μM. The maximal chloride currents at saturating glycine concentrations of 300 μM were not significantly different in $Glrb^{Eos/Eos}$ animals, despite a minor increase in the $EC_{50}$ ($Glrb^{WT/WT}$ 100 ± 5 μM, $Glrb^{Eos/Eos}$ 130 ± 9 μM, p=0.0123 t-test). In view of the millimolar concentration of glycine present during synaptic transmission (*Beato, 2008*; *Legendre, 1998*), these data indicate that the presence of the N-terminal fluorophore does not affect GlyR function under physiological conditions. Hill coefficients for $Glrb^{WT/WT}$ and $Glrb^{Eos/Eos}$ were in the range of 3.5–4, arguing for cooperativity of the subunits during glycine binding. The glycinergic origin of the chloride influx was confirmed by blocking the currents recorded in the presence of 100 μM glycine with 10 μM strychnine.

## Quantitative confocal imaging of endogenous mEos4b-GlyRβ and mRFP-gephyrin at spinal cord synapses in tissue

To verify the expression and synaptic targeting of the mEos4b-GlyRβ protein, we carried out quantitative confocal imaging in 40 μm vibratome tissue sections. $Glrb^{Eos/Eos}$ animals were crossed with a previously established KI mouse line expressing mRFP-tagged gephyrin to visualize inhibitory post-synaptic sites (*Specht et al., 2013*). Since the synaptic localization of the GlyR is strictly dependent on its interaction with gephyrin (*Patrizio et al., 2017*), we expected a high degree of co-localization of the two proteins in the brainstem and spinal cord (*Zeilhofer et al., 2005*). Indeed, mEos4b-GlyRβ was widely expressed at inhibitory synapses in the thalamus, midbrain, pons, and medulla (*Figure 1—figure supplement 3*). Very little fluorescence was detected in the forebrain despite the high reported expression of the *Glrb* transcript (*Fujita et al., 1991*; *Malosio et al., 1991*), suggesting that protein levels are controlled by post-transcriptional mechanisms in a region-specific manner, as previously proposed (*Weltzien et al., 2012*).

In the spinal cord, we observed bright punctate mEos4b-GlyRβ signals throughout the gray matter, with the exception of the superficial laminae of the dorsal horn, where the intensity of the green fluorescence was markedly lower (*Figure 1A*). The expression of mEos4b-GlyRβ and mRFP-gephyrin in homozygous and heterozygous animals was quantified in confocal images of thoracic and lumbar spinal cord slices at 2 months (*Figure 1B*) and 10 months of age (*Figure 1—figure supplement 4*). The same region of the ventral horn, indicated by the white square in *Figure 1A*, was taken for quantification of mEos4b-GlyRβ and mRFP-gephyrin expression in all conditions. The integrated mEos4b intensity at gephyrin-positive ventral horn synapses was exactly two times higher in $Glrb^{Eos/Eos}$ mice than in $Glrb^{Eos/WT}$, demonstrating that both alleles are expressed with the same efficiency, and that the mEos4b fluorophore does not affect the synaptic localization of the receptor complexes. To confirm that the confocal image acquisition was in the linear dynamic range, we bleached the mEos4b fluorophores

by repeatedly scanning the same tissue area at constant laser power (*Figure 1A*, white square), which resulted in a linear decay of pixel intensities over a range of more than 20-fold (*Figure 1C*).

Across the spinal cord slices, the intensity of synaptic mEos4b-GlyRβ puncta increased from dorsal to ventral both in homozygous (*Figure 1D and E*) and in heterozygous animals (*Figure 1—figure supplement 4*). Similarly, mRFP-gephyrin fluorescence was higher and more variable in the ventral horn, suggesting that synapses were on average about twice as big as those in the dorsal horn, despite being fewer in number (*Figure 1F* and *Figure 1—figure supplement 4*). The mEos4b/mRFP ratio was relatively equal across the spinal cord with the exception of the superficial layers of the dorsal horn, where GlyR levels were largely exceeded by gephyrin (*Figure 1A and D*). The lower GlyR-scaffold occupancy of synapses in laminae I–III can be explained by the predominant expression of $GABA_A$Rs that compete for receptor binding sites at these mixed inhibitory synapses (*Alvarez et al., 1996*; *Lorenzo et al., 2014*; *Todd et al., 1996*).

## Dual-color super-resolution imaging of glycinergic spinal cord synapses

To quantify the observed structural differences at super-resolution, we combined radial fluctuation (SRRF) analysis of mRFP-gephyrin and photo-activated localization microscopy (PALM, a form of SMLM) of mEos4b-GlyRβ in spinal cord tissue from double KI animals. Sucrose-impregnated cryo-sections of 2 µm thickness were prepared from dorsal and ventral tissue and placed on gridded coverslips (*Figure 2A*). SRRF and PALM images were acquired sequentially. First, mRFP signals were recorded with high-intensity 561 nm laser illumination until all mRFP fluorophores were bleached (10,000 frames). mEos4b was then photoconverted with increasing 405 nm laser intensity and imaged at 561 nm for 25,000 frames until all available fluorophores were exhausted. By acquiring both fluorophores, mRFP and photoconverted mEos4b, in the same emission band (607/36 nm), any chromatic misalignment between the two super-resolved images was eliminated. SRRF reconstruction was carried out on the raw mRFP image sequence and PALM images were generated from individual mEos4b detections using Gaussian peak fitting (*Figure 2B*). The spatial resolution was estimated using Fourier ring correlation (FRC), which measures the similarity of two images as a function of spatial frequency by comparing the odd and even frames of the raw image sequence. According to this analysis, the spatial resolution of SRRF was 46 nm and that of PALM 21 nm. It should be noted that the synaptic puncta in the SRRF images appear somewhat smaller and brighter due to differences in the reconstruction methods that result in differences in the dynamic intensity range.

The majority of synaptic clusters in the dual super-resolution images were small and spherical or elongated. Larger clusters displayed a variety of morphologies including elongated shapes seen in side view (*Figure 2B*) as well as convoluted structures, and were more frequently observed in the ventral horn (*Figure 2* and *Figure 2—figure supplement 1*). All mEos4b-GlyR clusters closely matched the mRFP-gephyrin clusters, confirming the localization of the receptors in the postsynaptic membrane. As expected of two directly interacting synaptic components, the degree of co-localization of mEos4b-GlyRβ and mRFP-gephyrin was very high, with mean intensity correlation quotients (ICQ) around 0.3 (*Figure 2C* and *Figure 2—figure supplement 1*), a value indicative of close spatial correlation (*Li et al., 2004*). Minor mismatches between the super-resolution images are explained by the fact that the majority of synapses are seen in cross-section and that the two fluorophores are located on opposite sides of the postsynaptic membrane (~30 nm distance; *Specht et al., 2013*). There were no obvious differences between the ICQ values of dorsal versus ventral synapses (0.28–0.3), indicating equivalent GlyRβ-gephyrin binding in the two regions. It should be noted that the ICQ reflects relative fluctuations between images and is not sensitive to absolute differences in signal intensities, resulting in similar ICQ values in animals that are heterozygous or homozygous for mRFP-gephyrin (*Figure 2C*). Quantitative comparison of mEos4b-GlyRβ and mRFP-gephyrin intensities confirmed that the amounts of receptor and scaffold proteins are closely correlated, and that the occupancy of receptor binding sites is the same in the dorsal and ventral horn, independent of synapse size (*Figure 2D* and *Figure 2—figure supplement 1*). To estimate the sizes of the synapses, we applied a density threshold to the PALM pointillist images and calculated the areas of the mEos4b-GlyRβ clusters (*Figure 2E*). The mean synapse area in the ventral horn was larger and more variable than in the dorsal region, both in animals of 2 and 10 months of age (*Figure 2F*). We also remarked that the overall number of synapses in ventral horn tissue was lower compared to the dorsal horn, significantly so by 10 months (*Figure 2G*; see also *Figure 1F* and *Figure 1—figure supplement 4*).

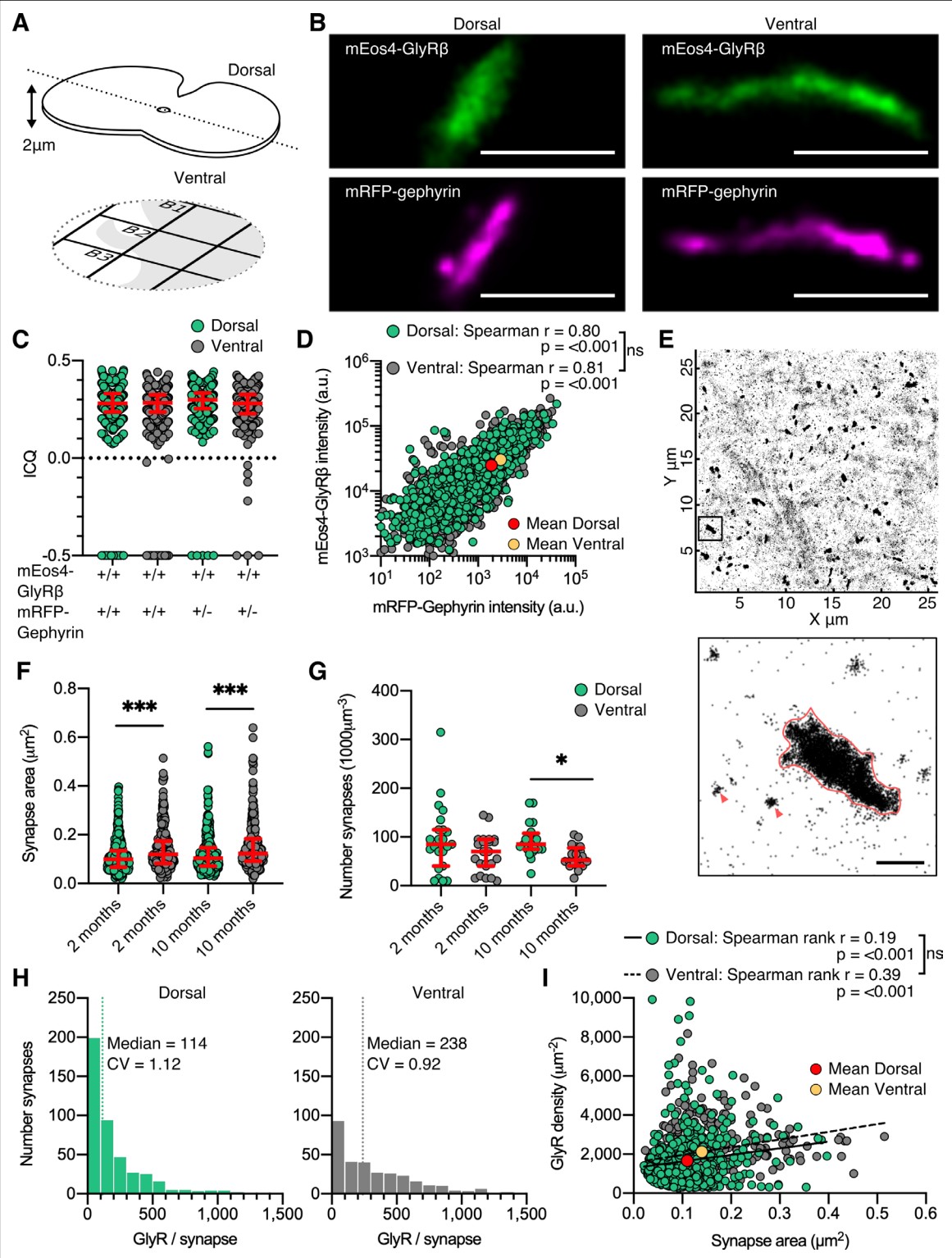

**Figure 2.** Dual-color super-resolution imaging of mEos4b-GlyRβ and mRFP-gephyrin. (**A**) 2 µm cryosections of spinal cord tissue were cut from dorsal and ventral tissue and placed on gridded glass coverslips. (**B**) Representative photo-activated localization microscopy (PALM) reconstruction of mEos4b-GlyRβ and super-resolution radial fluctuation (SRRF) reconstruction of mRFP-gephyrin at single dorsal and ventral synapses. Scale bar = 500 nm. (**C**) Intensity correlation quotient (ICQ) of mEos4b-GlyRβ and mRFP-gephyrin in heterozygous and homozygous 2-month-old mice. Plot shows median ± interquartile range. N = 357–604 synapses from 23 dorsal and 23 ventral images from 9 and 8 tissue slices, respectively, from 2 mice per condition. (**D**) Quantification of GlyR-gephyrin occupancy in 2-month-old homozygous mice (*Glrb*[Eos/Eos]/*Gphn*[mRFP/mRFP]). Nonparametric Spearman's rank shows the

*Figure 2 continued on next page*

*Figure 2 continued*

same positive correlation at dorsal and ventral synapses. N = 1115 dorsal synapses and 1107 ventral synapses. (**E**) Pointillist reconstruction of mEos4b-GlyRβ detections. Inset shows a single synapse; red arrows indicate extrasynaptic receptor complexes. Scale bar = 500 nm. (**F**) Area of dorsal vs. ventral synapses in 2- and 10-month-old homozygous mice. Plot shows median ± interquartile range. N = 234–433 synapses from 20 to 23 images from 7 to 9 tissue slices from 2 mice per age per condition. Nonparametric Kruskal–Wallis ANOVA with Dunn's multiple comparison test. (**G**) Number of synapses in dorsal and ventral tissue in 2- and 10-month-old homozygous mice. Plot shows median ± interquartile range. N = 20–23 images from 7 to 9 tissue slices from 2 mice per age per condition. Nonparametric Kruskal–Wallis ANOVA with Dunn's multiple comparison test. (**H**) Histogram of the number of glycine receptors (GlyRs) per synapse calculated from the molecular conversion of detections (see *Figure 2—figure supplements 3 and 4*) in 2-month-old homozygous mice. N = 433 dorsal synapses and 304 ventral synapses from 23 dorsal and 23 ventral images from 9 and 8 tissue slices, respectively, from 2 mice. CV, coefficient of variation. (**I**) Scatter plot of GlyR density vs. synapse area in 2-month-old homozygous mice shows no difference between dorsal and ventral synapse densities. N = 433 dorsal synapses and 304 ventral synapses. *p<0.05, **p<0.01, ***p<0.001.

The online version of this article includes the following figure supplement(s) for figure 2:

**Figure supplement 1.** Dual-color super-resolution synapse shape & 10 month correlation analysis.

**Figure supplement 2.** Quantification of mEos4b detections at synapses.

**Figure supplement 3.** Molecule conversion of mEos4b-GlyRβ detections into GlyRs copy numbers.

**Figure supplement 4.** Copy numbers and GlyRs densities at synapses (10 months).

The inverse relationship between synapse number and size is likely due to the presence of different cell types in the two regions, specifically Renshaw cells and large motor neurons in the ventral horn that require strong inhibitory control (e.g., *Bhumbra et al., 2014*).

## Quantification of GlyR numbers and densities at native spinal cord synapses

Given that mEos4b-GlyRβ subunits are expressed at endogenous levels in *Glrb*$^{Eos/Eos}$ animals and that all the fluorophores were exhausted during the PALM recordings, we were able to count the number of GlyRs at spinal cord synapses. The total number of mEos4b-GlyRβ detections at synapses (*Figure 2* and *Figure 2—figure supplement 2*) was converted into molecule numbers taking into account the blinking properties of the fluorophore and the $\alpha_3$:$\beta_2$ stoichiometry of the pentameric GlyR complex (*Durisic et al., 2014*; *Patrizio et al., 2017*). To this aim, the average number of detections per fluorophore (detections/burst) and the fraction of functional fluorophores (probability of detection, $P_{det}$; *Figure 2—figure supplement 3*) were determined in each set of experiments using extrasynaptic receptor complexes (*Figure 2E*, red arrowheads). We calculated a median copy number of 114 pentameric GlyR complexes at dorsal horn synapses and twice that number at ventral horn synapses in 2-month-old animals (*Figure 2H*). Copy numbers were almost identical at 10 months (*Figure 2—figure supplement 4*), indicating that the glycinergic network was mature at both time points. These numbers exceed estimates derived from electrophysiological recordings in newborn, juvenile, and adult rat spinal cord neurons that suggest the activation of as few as 7 and up to about 110 GlyRs during an average miniature inhibitory postsynaptic current (mIPSC) (*Chéry and de Koninck, 1999*; *Oleskevich et al., 1999*; *Singer and Berger, 1999*; *Takahashi, 1992*). The high numbers of GlyRs measured by fluorophore counting therefore imply that the available receptors are not saturated by quantal release, which is likely to increase the dynamic range of postsynaptic inhibition (*Alvarez, 2017*).

Our quantitative PALM data further demonstrate that differences in receptor numbers result from differences in synapse size (*Figure 2F*). By combining the two parameters, we derived mean GlyR densities of ~2000 μm$^{-2}$ (*Figure 2I* and *Figure 2—figure supplement 4*). Similar receptor densities of 1250 μm$^{-2}$ and ~2000 μm$^{-2}$ have been measured at GABAergic synapses in cerebellar stellate cells and in cultured hippocampal neurons, respectively (*Liu et al., 2020*; *Nusser et al., 1997*). We saw no differences in the GlyR packing density at dorsal and ventral horn synapses, nor did we find a clear size dependence, as determined by linear regression of all synapses (*Figure 2I*) and the evolution of the coefficient of variation of GlyR density with respect to synapse area (*Figure 2—figure supplement 4*). These findings are significant because they indicate that GlyR density is constant and largely independent of neuron type, embryonic origin, or physiological function. Assuming that gephyrin molecules are clustered at densities of up to 9000 μm$^{-2}$ (*Specht et al., 2013*), our data also suggest that close to 50% of the receptor binding sites are occupied by GlyRs at native spinal cord synapses, in line with

earlier observations of GlyR subunits that were overexpressed in cultured neurons (*Patrizio et al., 2017*).

## Quantitative SR-CLEM of GlyRβ molecular organization

To integrate the results of molecule counting with three-dimensional ultra-structural information and the exact synapse size, we further analyzed dorsal and ventral horn synapses by SR-CLEM. Previously imaged cryosections of *Glrb*[Eos/Eos] tissue from 10-month-old animals were embedded in epoxy resin, and ultra-thin (70 nm) serial sections were collected on electron microscopy (EM) slot grids with an ultramicrotome (*Figure 3A*). After osmium tetroxide enhancement, electron micrographs of identified synapses were acquired in all serial sections and registered manually using the coverslip grids and cellular structures as reference (*Figure 3B and C*). All of the synapses that were both imaged by PALM and reconstructed with EM were functionally mature as judged by the apposition of a single presynaptic terminal containing synaptic vesicles. In line with our PALM data, we found that glycinergic synapses in the ventral horn were substantially larger and more variable in size than those in the dorsal horn (*Figure 3C and D*). There was good correspondence between the calculated synapse areas in the EM and PALM image reconstructions, even though PALM underestimated the sizes of some large ventral horn synapses (*Figure 3—figure supplement 1*). This is probably due to the fact that a majority of synapses are tilted, and that the axial component of the area is not captured in the SMLM projections. Whereas most synapses in the dorsal horn were macular, ventral synapses were frequently composed of subdomains (*Figure 3E–G*). In agreement with earlier studies (*Alvarez et al., 1997*; *Lushnikova et al., 2011*; *Santuy et al., 2018*), the degree of complexity scales with the size of the postsynaptic specialization (*Figure 3F*), and was taken into account for the calculation of the combined area in the EM serial sections.

The ratio of GlyR copy numbers and the area of the inhibitory postsynaptic specialization obtained by EM resulted in average receptor densities of approximately 2000 μm$^{-2}$ (*Figure 3H*). Consistent with our PALM estimates, we did not observe significant differences between synapses in the dorsal and the ventral horn (*Figure 2I* and *Figure 2—figure supplement 4*). Furthermore, the GlyR packing density was not dependent on synapse size (*Figure 3I*), supporting an earlier proposal (*Lim et al., 1999*). This suggests that GlyRs are assembled in a systematic manner, where receptor numbers increase linearly with synapse size. Since the morphological complexity of synapses increases with size, it can also be concluded that GlyR occupancy at individual subdomains of the postsynaptic sites is uniform. GlyR densities were indeed not significantly different within subclusters of reconstructed synapses (*Figure 3G and I*, black data points). Together, these findings point to a tight regulation of the architecture of glycinergic synapses across different molecular length scales, where GlyRs are arranged in sub-synaptic signaling units.

## GlyR packing density is unaltered in the hyperekplexia mouse model *oscillator*

Having identified that GlyRs have a stereotypic molecular organization that is maintained throughout adulthood and across synapses in different neuronal circuits, we questioned whether this arrangement is altered in a mouse model of hyperekplexia, a motor-related neurological disease that significantly impacts motor processing in the ventral horn of the spinal cord. The mouse mutation *oscillator* (*Glra1*[spd-ot/spd-ot]) is recessively inherited and causes a microdeletion and frameshift in the TM3-4 intracellular loop of the GlyRα1 subunit, leading to subunit truncation and subsequent loss of functional GlyRs at synapses (*Kling et al., 1997*). Homozygous *oscillator* mice do not live past 3 weeks of age (*Buckwalter et al., 1994*). In contrast, heterozygous animals have a normal lifespan and exhibit a more subtle phenotype. *Glra1*[spd-ot/WT] mice display an increased startle reflex and lower GlyRα1 levels (*Kling et al., 1997*), making them a suitable model for human hyperekplexia. We generated mutant mice that were homozygous for mEos4b-GlyRβ (as described above) and heterozygous for *oscillator* (*Glrb*[Eos/Eos] / *Glra1*[spd-ot/WT]) as well as WT littermates (*Glrb*[Eos/Eos]/*Glra1*[WT/WT]). In these experiments, inhibitory synapses were detected by immunolabeling of endogenous gephyrin using the mAb7a antibody.

Quantitative confocal imaging in 40 μm vibratome tissue sections showed bright punctate mEos4b-GlyRβ signals localized at synapses (*Figure 4A*). Across the spinal cord slices, the intensity of synaptic mEos4b-GlyRβ puncta increased from dorsal to ventral in WT animals (*Figure 4B*), replicating the intensity profile observed previously (*Figure 1D*). The intensity of synaptic mEos4b-GlyRβ puncta in

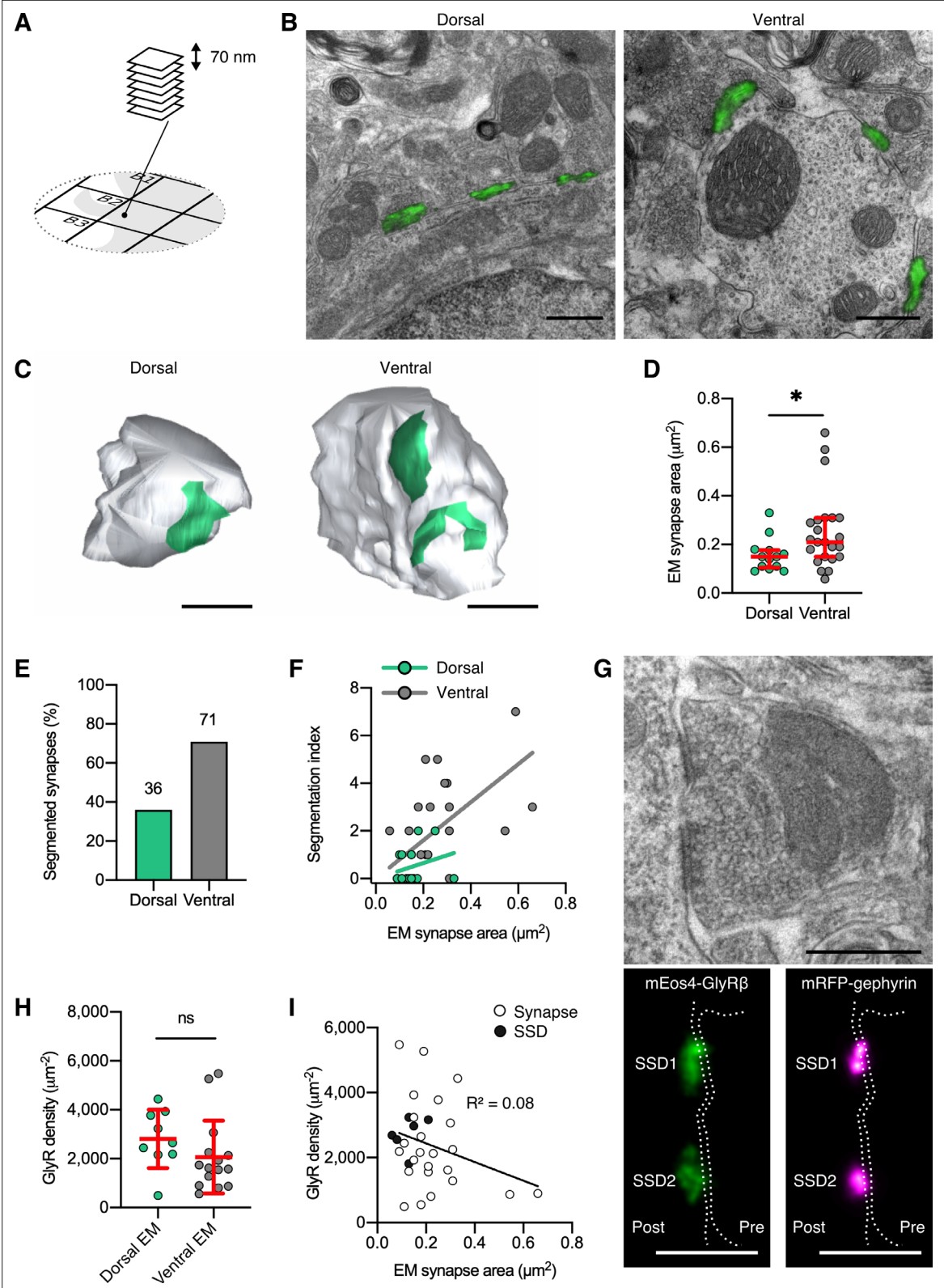

**Figure 3.** Quantitative super-resolution correlative light and electron microscopy (SR-CLEM) of endogenous glycine receptor (GlyR) molecular organization. (**A**) Samples used for photo-activated localization microscopy (PALM) were re-sectioned as serial ultra-thin 70 nm sections for CLEM. (**B**) Representative dorsal and ventral synapses imaged with PALM (mEos4b-GlyRβ; green) and super-imposed with their corresponding electron micrographs. Scale bar = 500 nm. (**C**) Representative 3D reconstructions of dorsal and ventral synapses. Green, postsynaptic site; gray, presynaptic

*Figure 3 continued on next page*

*Figure 3 continued*

bouton. Scale bar = 500 nm. (**D**) Area of dorsal and ventral synapses measured by EM. Plot shows median ± interquartile range. Nonparametric unpaired two-tailed t-test, Mann–Whitney post hoc. (**E**) Percentage of total synapses measured in EM with segmented shapes. (**F**) Comparison of segmentation index with synaptic area in dorsal and ventral synapses. (**G**) Juxtaposition of a raw electron micrograph and reconstructed PALM/super-resolution radial fluctuation (SRRF) images of sub-synaptic domains (SSDs) in the same ventral synapse. Scale bar = 500 nm. (**H**) Analysis of GlyR density following correction for EM area measurements. Plot shows mean ± SD. (**I**) Combined scatter plot of dorsal and ventral synapse densities shows density is independent of synapse size. White, all synapses; black, SSDs. N = 13 dorsal and 23 ventral synapses from 2 mice. *p<0.05, ns, not significant.

The online version of this article includes the following figure supplement(s) for figure 3:

**Figure supplement 1.** Comparison of PALM and EM area measurements.

heterozygous *oscillator* animals was substantially lower than those of WT littermates, which explains the reductions in membrane levels of GlyR and gephyrin previously observed by Western blotting (***Kling et al., 1997***). The mEos4b/gephyrin-7a ratio was relatively equal across the spinal cord with greater variation seen in *oscillator*. No ectopic GlyRβ clusters were detected, meaning that GlyRs and gephyrin always colocalized (***Figure 4—figure supplement 1***). The number of gephyrin-positive synapses across the spinal cord remained unchanged between WT and *oscillator*.

Using our quantitative PALM approach, we determined the number and size of glycinergic synapses in tissue slices of dorsal and ventral spinal cord, as well as the mEos4b detection density (***Figure 5*** and ***Figure 5—figure supplement 1***) in order to understand the alterations in glycinergic synapse architecture in this mutant mouse model. The detections per synapse were converted into molecule numbers as described before (***Figure 2—figure supplement 3*** and ***Figure 5—figure supplement 2***). In WT animals, we observed small and spherical dorsal synapses and larger, elongated ventral synapses, while *oscillator* synapses appeared small both in dorsal and ventral horn tissue (***Figure 5A***). This was confirmed by quantitative analysis. The mean synapse area in the ventral horn was significantly larger than in the dorsal region in WT animals (***Figure 5B***), in agreement with our earlier data (***Figure 2F***). However, this difference was lost in *Glra1*$^{\text{spd-ot/WT}}$ littermates. The overall number of synapses was lower in ventral horn tissue compared to the dorsal horn in both WT and *oscillator* animals (***Figure 5C***). These data suggest that the decrease in functional receptors in heterozygous *oscillator* mice manifests itself as a reduction in the size of ventral synapses.

We further quantified the total number of endogenous GlyRs at synapses (***Figure 5D***) and found that the number and distribution of receptors in WT mice matched that of 2- and 10-month-old WT animals analyzed previously (***Figure 2H*** and ***Figure 2—figure supplement 4***). This is remarkable considering that these experiments were performed independently and subsequent to the previous dataset, attesting to the stability of the measurements. It should also be pointed out that the GlyR copy numbers were indistinguishable from those in mRFP-gephyrin double KI animals, confirming that the presence of the fluorophores does not affect their expression and synaptic clustering. We found similar copy numbers in dorsal synapses of heterozygous *oscillator* mice compared to WT littermates. However, in ventral tissue, we found a shift towards lower receptor numbers per synapse, suggestive of smaller synapses. By combining the measurements of GlyR copy numbers with synapse area, we could derive the receptor density. Consistent with our previous PALM data, we found a constant receptor density independent of synapse size in WT mice, as judged by the shallow slope of the linear regression (***Figure 5E***). We observed a similar and constant GlyR packing density in synapses of *oscillator* mice. This suggests that despite a decrease in the total number of functional GlyRs in the heterozygous *oscillator* mouse model the molecular organization underlying receptor clustering within the synapse is maintained, in line with the lack of an overt neuromotor phenotype in these animals. In other words, the receptors are assembled stereotypically in the disease model, as in WT, with synapse size consistently scaling with receptor number. Together, our findings describe a highly regulated architecture of glycinergic synapses in both WT animals as well as in a model of synaptic pathology, providing a structural basis of glycinergic signaling.

## Discussion

Combining single-molecule PALM imaging, molecular counting, and 3D EM, we have shown that glycinergic synapses in different regions of the spinal cord follow the same structural principle insofar as their receptor-scaffold occupancy and packing densities are the same. This uniformity extends to

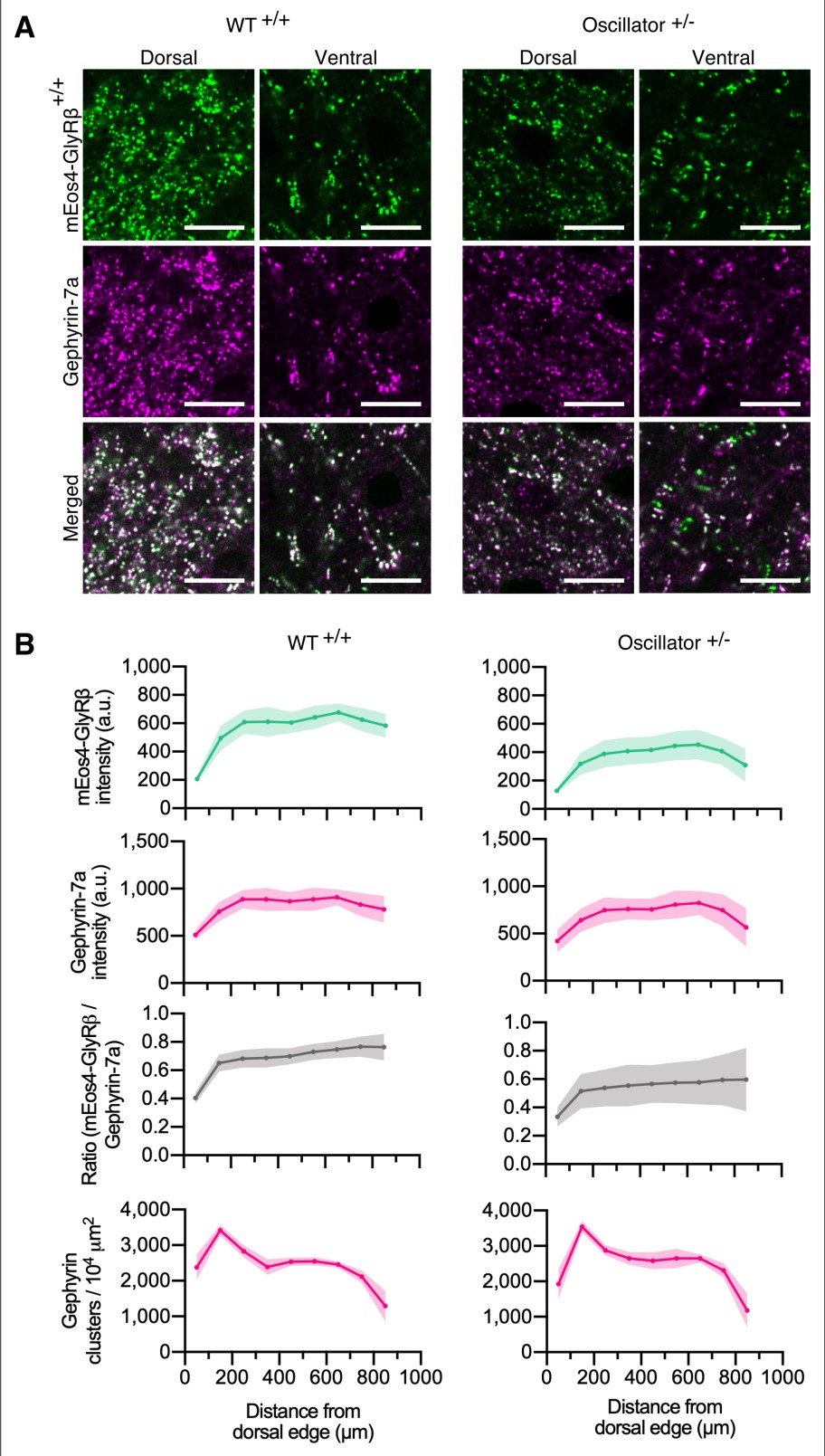

**Figure 4.** Quantitative confocal imaging of endogenous glycine receptors (GlyRs) in the *oscillator* mouse model. (**A**) Representative confocal images of dorsal and ventral synapses from heterozygous *oscillator* mice (+/-) compared to homozygous WT (+/+) littermates. All mice are homozygous for mEos4b-GlyRβ (green), with gephyrin-7a immunolabeling (magenta). Scale bar = 10 μm. (**B**) Mean intensity ± 95% confidence interval of

*Figure 4 continued on next page*

*Figure 4 continued*

mEos4b-GlyRβ and gephyrin-7a at gephyrin-positive puncta, and numbers of gephyrin clusters measured from distal edge of spinal cord in 2-month-old mice. N = 9–11 images from 9 to 11 tissue slices from 2 mice per genotype.

The online version of this article includes the following figure supplement(s) for figure 4:

**Figure supplement 1.** Quantitative confocal analysis at mEos4 puncta of the *oscillator* mouse model.

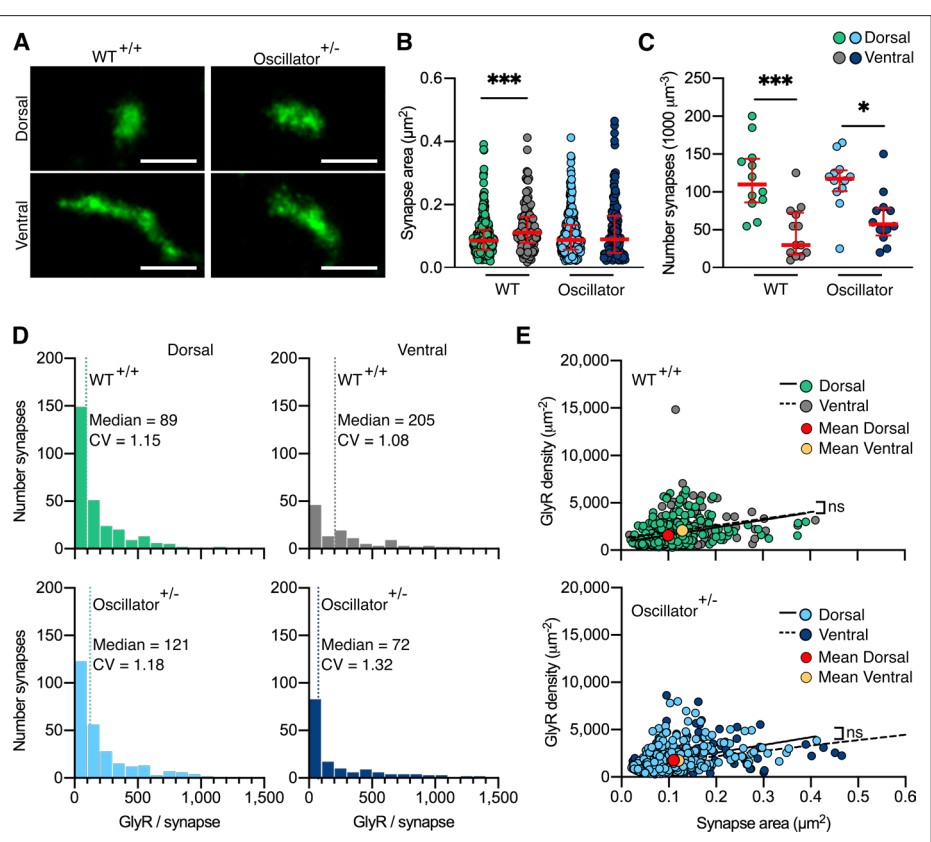

**Figure 5.** Photo-activated localization microscopy (PALM) imaging of the glycine receptor (GlyR) in the *oscillator* mouse model. (**A**) Representative PALM reconstructions of mEos4b-GlyRβ at single dorsal and ventral synapses in heterozygous (+/-) *oscillator* and homozygous (+/+) wild-type (WT) littermates. All mice are homozygous for mEos4b-GlyRβ. Scale bar = 500 nm. (**B**) Area of dorsal vs. ventral synapses in heterozygous *oscillator* vs. WT littermates. Plot shows median ± interquartile range. N = 120–282 synapses from 24 images from 9 to 11 tissue slices from 2 mice per genotype. Nonparametric Kruskal–Wallis ANOVA with Dunn's multiple comparison test. (**C**) Number of synapses in dorsal and ventral tissue in heterozygous *oscillator* vs. WT mice. Plot shows median ± interquartile range. N = 12–13 images from 2 mice per genotype. Parametric one-way ANOVA with Tukey's multiple comparison test. (**D**) Histogram of the number of GlyRs per synapse calculated from the molecular conversion of detections (see *Figure 2—figure supplement 3*, and *Figure 5—figure supplements 1 and 2*). N = 282 WT dorsal and 120 ventral synapses, 273 *oscillator* dorsal and 156 ventral synapses from 24 images from 9 to 11 tissue slices from 2 mice per genotype. CV, coefficient of variation. (**E**) Scatter plots of GlyR density vs. synapse area show no difference between dorsal and ventral synapse densities in WT and *oscillator*. N = same as in (**D**). *p<0.05, ***p<0.001, ns, not significant.

The online version of this article includes the following figure supplement(s) for figure 5:

**Figure supplement 1.** Quantification of mEos4b detections at synapses in the *oscillator* mouse model.

**Figure supplement 2.** Molecule conversion of mEos4b-GlyRβ detections into GlyRs copy numbers in the *oscillator* mouse model.

the sub-synaptic level. The presence of so-called sub-synaptic domains at inhibitory synapses has been shown by super-resolution microscopy (*Crosby et al., 2019*; *Dzyubenko et al., 2016*; *Pennacchietti et al., 2017*; *Specht et al., 2013*; *Yang et al., 2021*). However, it remains controversial whether the identified patterns represent the overall structure of the postsynaptic specialization itself or whether they reflect intra-synaptic variations in molecule clustering. Our quantitative SR-CLEM data lend support to the first model, whereby inhibitory postsynaptic sites in the spinal cord are composed of subdomains that determine the distribution of the GlyRs. This organization is achieved through direct interactions between GlyRs and gephyrin, as shown by the close correspondence between the receptors and scaffold proteins. As such, the GlyR subclusters at spinal cord synapses do not constitute SSDs within the postsynaptic membrane in the strict sense since they exhibit uniform binding to the synaptic scaffold (discussed in *Yang and Specht, 2019*). The stereotypic GlyR density within SSDs observed in our study supports the idea that these structures can instead be equated with the convolutions of the synaptic junction observed by EM (*Alvarez et al., 1997*; *Lushnikova et al., 2011*; *Peters and Palay, 1996*; *Santuy et al., 2018*). The formation of these convolutions is probably a consequence of gephyrin oligomerization that appears to introduce an asymmetry in the synaptic scaffold. The situation may be different at GABAergic synapses, where the coexistence of gephyrin-dependent and gephyrin-independent clustering mechanisms could lead to the formation of spatially more restricted SSDs containing different GABA$_A$R subtypes (*Pennacchietti et al., 2017*; *Specht, 2020*).

The nanoscale organization of inhibitory synapses is the same in glycinergic neurons in the dorsal and the ventral spinal cord, despite their different embryonic origins. Ventral horn synapses are generally larger, more complex, and contain more GlyRs, suggesting that the size of the synapse is differentially regulated in a regional and cell-type-specific manner to adjust the level of glycinergic inhibition. As such, these synapses may be particularly well adapted to motor circuits, assuring strong and reliable inhibition of the postsynaptic neuron (*Alvarez, 2017*). At the same time, the long and tortuous perimeter of the postsynaptic specialization is likely to accelerate the dynamic exchange of GlyRs and other synaptic components (*Chow et al., 2017*; *Santuy et al., 2018*), thereby promoting the molecular plasticity at complex inhibitory synapses (*Specht, 2020*). Whether glycinergic plasticity results from transient changes in GlyR occupancy and/or from the recruitment of extrasynaptic GlyR-gephyrin complexes (*Chapdelaine et al., 2021*) has not been proven thus far. However, our data suggest that ultimately it is the size and complexity rather than the GlyR packing density that is dynamically regulated. It can further be argued that large and morphologically complex synapses may be particularly well adapted to integrate fast repetitive, or indeed multivesicular release arising from one or more presynaptic sites, thus providing a strong and reliable inhibition of the postsynaptic neuron while maintaining fast neurotransmitter clearance (discussed in *Alvarez, 2017*; *Rudolph et al., 2015*).

In addition, our data show that GlyR density and occupancy do not change between 2 and 10 months of age, indicating that receptor clustering is fully mature by the earlier time point. Studies of normal aging of spinal cord synapses are scarce, and its effect on receptor organization has not been studied. Broadhead and colleagues (*Broadhead et al., 2020*) report no difference between the number of excitatory synapses in the ventral horn and only a slight increase in dorsal synapses between 2- and 9-month-old mice. Broadly in line with these findings, we found no difference in the number of synapses, synapse area, and GlyR packing density in dorsal and ventral tissue between 2 and 10 months. Thus, glycinergic postsynaptic sites show considerable control over their molecular composition throughout adulthood, further emphasizing the functional significance of their synaptic architecture in both sensory and motor signaling. Our data therefore suggest that a constant GlyR density potentially provides the most efficient organization of the glycinergic postsynaptic site, while enabling the refinement of the size and complexity of the synapse due to ongoing neural activity.

Their molecular organization sets glycinergic synapses apart from excitatory synapses that do not exhibit systematic receptor clustering. Different glutamate receptors are highly variable and occupy separate SSDs within the overall postsynaptic density (PSD) (*Goncalves et al., 2020*). The number of AMPARs at excitatory synapses can range from essentially 0 (at silent synapses) to more than 100 (*Nusser et al., 1998*). Within SSDs, the average AMPAR density has been estimated at around 5000 µm$^{-2}$ (*Goncalves et al., 2020*), although the absolute values will have to be confirmed using quantitative labeling strategies such as the one described here. Interestingly, AMPAR content of SSDs was shown to vary in response to synaptic plasticity without an apparent change in SSD size (*Compans*

*et al., 2021*). This suggests that the packing density of AMPARs is not constant even at the nanoscale, which may be the basis for AMPAR plasticity at excitatory synapses.

The neuromotor disease hyperekplexia results from defects in glycinergic inhibition in humans. Several mouse models with analogous mutations in the *Glra1* gene recapitulate the phenotype of exaggerated startle reflexes and muscle stiffness (*Schaefer et al., 2018*). In general, the mutations in the various mouse models are less well tolerated than in humans and often have lethal phenotypes in homozygotes. Hyperekplexia can be recessively or dominantly inherited in humans, but neither has been shown to cause lethality. The *oscillator* mouse model, whilst lethal in the homozygous form, displays a relatively mild phenotype in heterozygous animals, with a measurable startle reflex and normal lifespan (*Kling et al., 1997*). Heterozygous *oscillator* therefore represents a relevant model for the subtler phenotype in humans and the long-term stability of glycinergic synapses beyond the developmental stage at which lethality occurs in homozygous animals. However, it is not known how the reduced GlyR levels in the $Glra1^{spd-ot/WT}$ hypomorph can affect (and sustain) functional motor networks. Our characterization of the molecular organization of heterozygous *oscillator* synapses shows that GlyR packing follows the same principle as in WT synapses, even though the total number of available functional receptors is reduced, resulting in smaller synapses in the ventral spinal cord. This further emphasizes that the stereotypic arrangement of GlyRs dictates the size of the postsynaptic domain. Most of the synapses that are formed in heterozygous *oscillator* likely achieve a size threshold capable of sustaining glycinergic signaling without serious motor defects. The lack of fundamental structural changes at glycinergic synapses further suggests that no or only limited compensatory effects take place in *oscillator*, in agreement with earlier findings that α1β heteropentameric GlyR complexes cannot be compensated for by other subunit configurations, α1-homopentamers, or GABA$_A$Rs (*Schaefer et al., 2012*). Our findings thus provide a new perspective into the molecular basis of GlyRα1 deficiency in an animal model of human hyperekplexia.

Taken together, our data show that dorsal and ventral synapses are distinct populations. Ventral horn synapses have much higher GlyR copy numbers, even though receptor density is not different. In contrast to the relatively compact, macular synapses in the dorsal horn, ventral horn synapses achieve a greater receptor number by enlarging the synaptic surface, thus multiplying the sites of signal transmission. These region-specific glycinergic synapse morphologies are likely to underlie functional differences at sensory (dorsal) versus motor (ventral) circuits.

## Materials and methods

### KI mouse model generation

The KI mouse line C57BL/6N-$Glrb^{tm1lcs}$ (MGI:6331106) carrying the mutant allele $Glrb^{tm1(Eos4)lcs}$ (MGI:6331065) was created by homologous recombination at the Institut Clinique de la Souris (ICS). Flanked by 5′ and 3′ homology arms of 1.23 kb and 3.49 kb, respectively, the targeting vector encompassed exon 2 of the *Glrb* gene with an insertion of the coding sequence of mEos4b, as well as a *floxed* neomycin selection cassette containing the Cre recombinase under control of protamine promoter in introns 2–3. The selection cassette was excised in the F1 generation by germline expression of Cre, resulting in a single *loxP* site in introns 2–3 of the *Glrb* locus (*Figure 1—figure supplement 1A*). The correct insertion of the mEos4b coding sequence was confirmed by sequencing of genomic $Glrb^{Eos/Eos}$ tail DNA. Genotyping was done using three primers (primer 1: TACCTTCTTGTTTTCTCTCC; primer 2: GTCTGTTTTCCCTCATAAGG; primer 3: TCGCTTTTGTAAATGATATGG) for the amplification of the mutant $Glrb^{Eos}$ (243 bp product) and/or the WT alleles (404 bp).

Purified spinal cord mRNA of $Glrb^{Eos/Eos}$, $Glrb^{Eos/WT}$, and $Glrb^{WT/WT}$ animals was reverse transcribed (primer 6: GGAGTCTAACAGTAATCTGG), and amplified (primer 4: AGGCGCGTCAAACTCGG; primer 5: CCATACCAACCAATGAAAGG). The correct splicing of the mutant transcript was confirmed by sequencing of amplified cDNA. For semi-quantitative RT-PCR, the $Glrb^{Eos/Eos}$ mRNA was spiked with WT mRNA at a ratio of 1:2, 1:1, and 2:1 and amplified (*Figure 1—figure supplement 1B*).

All experiments (with the exception of the data in *Figures 4 and 5*, *Figure 1—figure supplements 1 and 2*, *Figure 4—figure supplement 1*, and *Figure 5—figure supplements 1 and 2*) were carried out with F2 animals resulting from a cross between the KI line C57BL/6N-$Glrb^{tm1lcs}$ (see above) and a KI mouse line expressing mRFP-tagged gephyrin ($Gphn^{mRFP}$) (*Machado et al., 2011*) in the C57BL/6J strain, meaning that the mice had a mixed C57BL/6N × C57BL/6J genetic background.

Adult *Glra1*[spd-ot] mice (*oscillator*, JAX stock #000536, RRID:IMSR_JAX:000536) from Jackson Laboratories (Bar Harbor, ME) were transferred to the animal facility of the Institute for Clinical Neurobiology (Würzburg, Germany). Genotyping was done using primer 7: GCCTCCGTGCTTTCTCCCTGC and primer 8: CCAGCCACGCCCCAAAG for the amplification of the mutant *Glra1*[spt-ot] (187 bp product) and/or the WT alleles (194 bp). *Oscillator* mice were backcrossed into the C57BL/6J background for at least 15 generations. Heterozygous *Glrb*[Eos/WT] animals were crossed with heterozygous *oscillator* mice for two generations, giving rise to F2 heterozygous *oscillator* animals that are homozygous for the *Glrb*[tm1(Eos4)Ics] allele (*Glra*1[+/spd-ot]/*Glrb*[Eos/Eos]). These animals had a mixed C57BL/6N × C57BL/6J genetic background and were used for the experiments shown in *Figures 4 and 5*, *Figure 4—figure supplement 1*, and *Figure 5—figure supplements 1 and 2*.

All experiments were in accordance with the European Union guidelines and approved by the local veterinary authorities. Animals at IBENS were treated in accordance with the guidelines of the French Ministry of Agriculture and Direction Départementale des Services Vétérinaires de Paris (École Normale Supérieure, Animalerie des Rongeurs, license B 75-05-20). Procedures carried out at the Institute for Clinical Neurobiology were approved by the Veterinäramt der Stadt Würzburg and the Committee on the Ethics of Animal Experiments (Regierung von Unterfranken, Würzburg) and authorized under reference numbers 55.2-2531.01-09/14; 55.2.2-2532.2-949-31.

## Primary cultures of spinal cord neurons

Cultures of mixed spinal cord neurons were prepared at embryonic day 13 (E13) from C57BL/6J WT *Glrb*[WT/WT] and *Glrb*[Eos/Eos] littermates. Spinal cord tissue was trypsinized in trypsin/EDTA (1 mg/ml) and DNase I (final concentration 0.1 mg/ml) for 20 min at 37°C. Trypsinization was stopped with 10% FCS. After trituration, spinal cord neurons were centrifuged at 800 rpm for 10 min. Cells were plated in a 3 cm dish on polylysine-coated coverslips at a density of 2–2.5 × 10⁵ cells/dish. Neurons were kept at 37°C and 5% $CO_2$ in neurobasal medium containing 2 mM L-glutamine and B27 supplement (Thermo Fisher Scientific) with an exchange of half the medium after 4 days in culture.

## Electrophysiological recordings

Spinal cord neuronal cultures at day in vitro 13 (DIV13) were used for patch-clamp recordings in whole-cell configuration. Currents were amplified with a EPC-10 amplifier (HEKA). A laminar flow of increasing agonist concentration (1, 10, 30, 60, 100, and 300 µM glycine, and 100 µM glycine/10 µM strychnine) was applied to the suspended cell using an Octaflow II system (ALA Scientific Instruments), allowing 10–30 ms for equilibration. The external buffer for spinal cord neurons was (in mM) 142 NaCl, 8.1 KCl, 1 $CaCl_2$, 6 $MgCl_2$, 10 glucose, 10 HEPES, pH adjusted to 7.4 with NaOH. To block neuronal excitability and ligand-gated ion channels, the external buffer was complemented with 1 µM TTX, 10 µM bicuculline, 10 µM CNQX, and 25 µM AP-5. The internal buffer was (in mM) 153 KCl, 1 $MgCl_2$, 5 EGTA, 10 HEPES, pH adjusted to 7.4 with CsOH. Recording pipettes were fabricated from borosilicate capillaries with an open resistance of 4–6 MΩ. Currents were measured at a holding potential of –70 mV. All experiments were performed at 22°C. The mean current at each glycine concentration was determined from the peak current amplitudes measured in N = 10–11 cells per genotype from three independent preparations (biological replicates).

## Spinal cord and brain tissue preparation and vibratome slices

Mice were sacrificed at 2 and 10 months of age by perfusion with 4% w/v paraformaldehyde (PFA; Polysciences, EM grade) and 0.1% v/v glutaraldehyde (GA; CliniSciences) in phosphate buffered saline (PBS, pH 7.4). Perfused animals were kept on ice for 30 min, followed by the dissection of the brain and spinal cord in PBS. Tissue was post-fixed in 4% w/v PFA in PBS overnight at 4°C. Brain and spinal cord tissue was rinsed in PBS, cut into smaller segments of thoracic and lumbar regions of the spinal cord and sliced on a vibratome (Leica) at a thickness of 40 µm (for confocal imaging) and 300 µm (for Tokuyasu preparation), and stored in PBS at 4°C.

## Confocal imaging and analysis

In order to label neuronal cells in brain slices (*Figure 1—figure supplement 3*) and inhibitory synapses in *oscillator* and WT littermates (*Figure 4*), free-floating vibratome slices (40 µm thickness) were blocked and permeabilized in PBS containing 0.25% Triton X100 (Sigma) and 0.1% fish gelatin

(Sigma) for 1 hr, and immunolabeled with either a primary antibody against NeuN (guinea pig polyclonal, 1:500 dilution, Millipore, #ABN90, RRID:AB_11205592) or gephyrin (mouse monoclonal, mAb7a, 1:500 dilution, Synaptic Systems, #147011, RRID:AB_887717) in PBS containing 0.1% Triton X100 and 0.1% fish gelatin overnight, followed by 3 hr incubation with Alexa Fluor 647-conjugated secondary antibody (donkey anti-guinea pig, 1:1000 dilution, Jackson ImmunoResearch, #706-605-148, RRID:AB_2340476) or Cy3-conjugated secondary antibody (goat anti mouse, 1:1000 dilution, Jackson ImmunoResearch, #115-165-003, RRID:AB_2338680) respectively.

Glass slides (Vector Laboratories) were cleaned with 70% v/v ethanol (Sigma), and vibratome sections were rinsed three times in PBS and mounted onto the glass slides. The glass slides were then briefly rinsed in distilled water and dried. A drop of VectaShield (Vector Laboratories) was added to each spinal cord section and covered with a #1.5 glass coverslip, which was sealed with PicoDent Twinsil Speed (equal weights of catalyst and base). Slides were stored at 4°C for confocal imaging.

Confocal imaging was carried out on a Leica SP8 TCX microscope using a Leica HC PL APO ×40/1.30 NA oil-immersion objective (Leica) and captured in 8-Bit using the Leica LAS-X software (RRID:SCR_013673) with setting HyD3. Images were captured sequentially, with laser illumination at wavelength 570 nm (mRFP, Cy3) imaged first, followed by laser illumination at 491 nm wavelength (mEos4). A cross-section from the dorsal horn to the ventral horn was imaged at a zoom of 5, speed of 25, 512 × 512 pixel (px) format. For decay analysis, eight consecutive frames were captured at a zoom of 5, speed 25, 512 × 512 px format. To tile the whole spinal cord, images were captured in at a zoom of 1, speed 100, 256 × 256 px format.

To ensure alignment of the clusters for the decay traces, images were opened in the image analysis software ICY (RRID:SCR_010587), and the rigid registration plug-in used, taking the first frame of mRFP-gephyrin as reference. The mRFP-gephyrin/ and mEos4b-GlyRβ channels were then separated and the Spot Detector plug-in *de Chaumont et al., 2012* used to identify the clusters in each frame in the mRFP-gephyrin channel, with the identified clusters saved as a region of interest (ROI) set. Using the image analysis software Fiji (RRID:SCR_002285), the identified mRFP-gephyrin-positive cluster ROI-Set was used to identify inhibitory synapses in the first frame of the mEos4b-GlyRβ channel. These inhibitory synapses were binned based on mEos4b intensity gray levels (5–12, 13–24, 25–49, 50–74, 75–99, 100–124, 125–255) in frame 1 and a new ROI-Set generated for each bin. Using the frame 1 intensity ROI-Sets, the integrated intensity of mEos4b was then measured at individual clusters across the eight frames. This enabled decay analysis of mEos4b intensity at synapses relative to their starting intensity (see *Figure 1C*).

In order to analyze the intensity of mRFP-gephyrin and mEos4b-GlyRβ clusters within the spinal cords from mice of different genotypes, the identified mRFP-gephyrin clusters from the first frame of the decay traces (as measured by the ICY Spot Detector plug-in, see above) were used to measure the relative intensity of mRFP-gephyrin and mEos4b-GlyRβ clusters at those locations. The ROI-Set of all mRFP-gephyrin-positive clusters was used in Fiji to identify inhibitory synapses, where the integrated intensity of mRFP and mEos4b was measured for each synapse (*Figure 1B*).

For the cross-sectional analysis, the mRFP-gephyrin/gephyrin-7a clusters were identified across the imaged tissue using the ICY Spot Detector plug-in, as described above, and saved as an ROI-Set. In Fiji, the integrated intensity of these identified clusters was measured in the mRFP-gephyrin/gephyrin-7a channel and the mEos4b-GlyRβ channel (*Figure 1D and F*).

## Cryosectioning of sucrose impregnated spinal cord tissue

Sucrose impregnated cryosections were prepared using an ultracryotomy protocol adapted from *Tokuyasu, 1973*. The 300 μm spinal cord vibratome slices were transferred into a 2.3 M sucrose solution in PBS overnight at 4°C and micro-dissected to isolate gray matter of the dorsal and the ventral horn region. These fragments were placed individually on top of drops of sucrose solution on aluminum EM pins (Leica) and immediately frozen in liquid nitrogen. Sections of 2 μm thickness were sliced on an ultramicrotome (Leica EM UC6) at –80°C and placed onto gridded coverslips (type 1.5H, ibidi GmbH), covered in PBS, and stored at 4°C for a maximum of 5 days before imaging.

## Single-molecule localization microscopy

Sucrose cryosections on gridded coverslips were rinsed once in PBS and imaged in PBS. Dual-color super-resolution images were acquired on an inverted Nikon Eclipse Ti microscope with a ×100/1.49

NA oil-immersion objective, with an additional 1.5× lens in the emission path, using an Andor iXon EMCCD camera (16-Bit, 107 nm pixel size), and NIS-Elements software (Nikon, RRID:SCR_014329). An emission filter 607/36 was chosen for imaging both mRFP-gephyrin and mEos4b-GlyRβ. Brightfield images were taken of the whole grid square identifying tissue structures. Lamp images were taken of the unconverted mEos4b-GlyRβ and mRFP-gephyrin (10 frames of 100 ms, ND8). mRFP-gephyrin movies of 10,000 frames were recorded with HiLo 561 nm continuous laser illumination (output power 50%, 400 mW, 50 ms frames). This was followed by 2 min of 100% 561 nm laser illumination to ensure all mRFP-gephyrin was bleached. Movies of 25,000 frames were recorded with photoconversion of mEos4b-GlyRβ by 0.5 ms pulsed 405 nm laser illumination (gradually increased to 100% by frame 22,000) with continuous 561 nm laser illumination (output power 50%, 400 mW, 50 ms frames). The focal plane was maintained using a Nikon perfect focus system.

## SMLM image analysis (SRRF and PALM)

Frames 100–6000 of the mRFP-gephyrin movies were taken for analysis (to remove saturated frames at the beginning and bleached frames at the end) and were drift corrected and reconstructed using NanoJ-SRRF plug-in for Fiji (*Gustafsson et al., 2016*).

Quantification of mEos4b-GlyRβ was carried out using a lab script for MATLAB (MathWorks, RRID:SCR_001622). The mEos4b single fluorophores were detected by Gaussian fitting. The resulting pointillist images were drift corrected in the x/y plane using five dense clusters of detections over a sliding window of 2000 frames. Rendered images were produced with a pixel size of 10 nm, sigma 0.01.

FRC was used to estimate the spatial resolution by dividing the odd and even frames of a raw mRFP-gephyrin movie and a raw mEos4b-GlyRβ movie and analyzing the resulting image stacks by SRRF and PALM, respectively. The images reconstructed from the odd and even frames were then compared using the FRC tool of the NanoJ-SQUIRREL plug-in for Fiji (*Culley et al., 2018*).

The mRFP-gephyrin and mEos4b-GlyRβ-rendered images were aligned by rigid registration using the Fiji plug-in TurboReg. The co-localization of mRFP-gephyrin and mEos4b-GlyRβ was carried out by individually cropping each synapse as separate images. The Fiji plug-in ICQ was then applied to each synapse (*Li et al., 2004*). The occupancy analysis was analyzed by thresholding the synapses in the mEos4b-GlyRβ images and measuring the intensity of each synapse in both channels. To analyze the PALM mEos4b-GlyRβ clusters, a lab written script for MATLAB (CountMol; *Patrizio et al., 2017*) was used to identify synapses (minimum number of detections 250, minimum cluster size 200 nm, maximum cluster size 3000 nm) and an intensity threshold of 0.1. For molecule conversion, CountMol was used to identify extrasynaptic receptor complexes (minimum number of detections 5, minimum cluster size 10 nm, maximum cluster size 120 nm). The number of detections per burst (identified as a minimum of two detections, with one burst per 1,000 frames) and the probability of detection $P_{det} = \frac{2}{N_1/N_2+2}$ were calculated and used to convert the detections to mEos4b-GlyRβ molecules (*Durisic et al., 2014*; *Patrizio et al., 2017*); see *Figure 2—figure supplement 3*, and *Figure 5—figure supplement 2*.

## Electron microscopy

Cryosections used for SMLM imaging on gridded coverslips were postfixed by incubation in 1% osmium tetroxide for 1 hr at 4°C, dehydrated in graded ethanol concentrations, and embedded in araldite epoxy resin. Grid squares imaged in SMLM were identified using the grid pattern imprinted in the resin. Serial ultra-thin 70 nm sections of these regions were cut and transferred onto formvar-coated EM grids (0.432 mm slot grids) using a UC6 ultramicrotome (Leica). Sections were counter-stained with 5% uranyl acetate in 70% methanol for 10 min, then washed in distilled water and air-dried before observation on a Philips TECNAI 12 microscope (Thermo Fisher Scientific).

For 3D synapse reconstruction, synapses were manually outlined in each serial section image using Fiji, followed by manual rotation and coarse alignment using the software GIMP (RRID:SCR_003182), then fine alignment of the synaptic area with the Microscopy Image Browser (MIB) software (RRID:SCR_016560). The aligned images were then opened in IMOD software (RRID:SCR_003297) to generate the 3D reconstruction.

The length of the synaptic junction of identified synapses was measured in high-magnification EM images with Fiji software. The total area of the postsynaptic surface was calculated as the cumulative

length of the postsynaptic domain in the entire stack of serial sections multiplied by the thickness of each section (70 nm). The segmentation index corresponds to the number of gaps in the postsynaptic site that were detected in the x/y plane of the images or along the z-axis (i.e., an interruption of the postsynaptic site in one or several continuous sections in the stack) and represents an estimate of the morphological complexity of the synapse.

## Graphing and statistical analysis

All graphing and statistical analysis were carried out using the software GraphPad Prism (RRID:SCR_002798) v.8 for all except the electrophysiology experiments that were carried out in v.9. Data were tested for normality of distribution using D'Agostino–Pearson and Kolmogorov–Smirnov tests. Data are represented as dot plots with median ± interquartile range (IQR), or histograms, unless otherwise stated. $*p<0.05$, $**p<0.01$, $***p<0.001$, ns, not significant.

## Acknowledgements

SAM was supported by a Fondation pour la Recherche Médicale (FRM) postdoctoral fellowship (SPF201809007132). NS was supported by funds of the Bavarian State Ministry of Science and the Arts and the University of Würzburg to the Graduate School of Life Sciences (GSLS), University of Würzburg. Research in our laboratory at IBENS was funded by the European Research Council (ERC, Plastinhib and Microcops), Agence Nationale de la Recherche (ANR, Synaptune and Syntrack), Labex (Memolife) and France-BioImaging (FBI). CV was supported by the Deutsche Forschungsgemeinschaft (DFG, VI586). We acknowledge the use of the EM platform of IBENS. We thank Pascal Legendre (ENP, Inserm) for insightful discussions and Constant Morez and Nadine Schibille for helpful comments on the manuscript. We also thank Marie-Christine Birling and Eve Geronimus from the Institut Clinique de la Souris (ICS, Illkirch, France) for the generation of the C57BL/6N-*Glrb*[tm1lcs] mouse line.

## Additional information

### Funding

| Funder | Grant reference number | Author |
|---|---|---|
| H2020 European Research Council | Plastinhib | Antoine Triller |
| Agence Nationale de la Recherche | Synaptune | Antoine Triller |
| Agence Nationale de la Recherche | Syntrack | Antoine Triller |
| Labex | Memolife | Antoine Triller |
| France Bio-Imaging | | Antoine Triller |
| Deutsche Forschungsgemeinschaft | VI586 | Carmen Villmann |
| Fondation pour la Recherche Médicale | SPF201809007132 | Stephanie Maynard |
| Bavarian State Ministry of Science and the Arts and the University of Würzburg | Graduate School of Life Sciences (GSLS) | Natascha Schaefer |
| European Research Council | MICRO-COPS | Antoine Triller |

The funders had no role in study design, data collection and interpretation, or the decision to submit the work for publication.

## Author contributions
Stephanie A Maynard, Formal analysis, Writing – original draft, Writing – review and editing, Conceptualization, Funding acquisition, Investigation; Philippe Rostaing, Formal analysis, Investigation; Natascha Schaefer, Formal analysis, Investigation, Funding acquisition; Olivier Gemin, Adrien Candat, Andréa Dumoulin, Investigation; Carmen Villmann, Formal analysis, Funding acquisition, Investigation, Resources; Antoine Triller, Funding acquisition; Christian G Specht, Conceptualization, Formal analysis, Investigation, Methodology, Writing – original draft, Writing – review and editing

## Author ORCIDs
Stephanie A Maynard ⓘ http://orcid.org/0000-0002-7838-3676
Natascha Schaefer ⓘ http://orcid.org/0000-0001-9743-1963
Olivier Gemin ⓘ http://orcid.org/0000-0003-3210-7876
Carmen Villmann ⓘ http://orcid.org/0000-0003-1498-6950
Antoine Triller ⓘ http://orcid.org/0000-0002-7530-1233
Christian G Specht ⓘ http://orcid.org/0000-0001-6038-7735

## Ethics
All experiments were in accordance with European Union guidelines and approved by the local veterinary authorities. Animals at IBENS were treated in accordance with the guidelines of the French Ministry of Agriculture and Direction Départementale des Services Vétérinaires de Paris (École Normale Supérieure, Animalerie des Rongeurs, license B 75-05-20). Procedures carried out at the Institute for Clinical Neurobiology were approved by the Veterinäramt der Stadt Würzburg and the Committee on the Ethics of Animal Experiments (Regierung von Unterfranken, Würzburg) and authorized under reference numbers 55.2-2531.01-09/14; 55.2.2-2532.2-949-31.

## Decision letter and Author response
Decision letter https://doi.org/10.7554/eLife.74441.sa1
Author response https://doi.org/10.7554/eLife.74441.sa2

# Additional files

## Supplementary files
• Transparent reporting form

## Data availability
All data generated and analyzed during this study are included in the manuscript and supporting files. The raw data are available for download from the online Figshare repository at https://doi.org/10.6084/m9.figshare.17816747.v1 or via the online repository at https://www.opendata.bio.ens.psl.eu/Super-CLEM-GlyR/.

The following dataset was generated:

| Author(s) | Year | Dataset title | Dataset URL | Database and Identifier |
|---|---|---|---|---|
| Maynard SA, Rostaing P, Schaefer N, Gemin O, Candat A, Dumoulin A, Villmann C, Triller A, Specht CG | 2022 | Identification of a stereotypic molecular arrangement of endogenous glycine receptors at spinal cord synapses | https://doi.org/10.6084/m9.figshare.17816747.v1 | figshare, 10.6084/m9.figshare.17816747.v1 |

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
