## [Editor Report]

The manuscript presents a quantitative and comprehensive characterization of glycinergic synapses and spinal cord synapse organization that will be a highly valuable resource for the field. The use of cutting-edge imaging and quantitative techniques allows an understanding of these understudied synapses at a high level.

---

## [Decision Letter]

[Editors' note: this paper was reviewed by Review Commons.]

---

## [Author Response]

Reviewer 1:In this manuscript Maynard et al. describe a newly generated knockin mouse to study the endogenous distribution of Gly receptors in the spinal cord. Using quantitative confocal imaging and SMLM the distribution and levels of GlyRs at spinal cord synapses is compared between dorsal and ventral horn. They found that levels of synaptic GlyR are higher in dorsal than ventral spinal cord synapses. Nevertheless, the ratio to gephyrin seems constant, except for synapses in superficial layers of the dorsal horn, where gephyrin levels exceeded the levels of GlyRs. There are also fewer, but larger synapses in the ventral horn than in the dorsal horn. These findings are further corroborated by an SR-CLEM approach. Furthermore, it is shown that in a mouse model for hyperekplexia GlyR levels are lower, but still enriched at synapses, and the dorsal-ventral gradient in GlyR expression was maintained. The difference in size of ventral and dorsal synapses observed in WT animals was also lost in the oscillator mouse, suggesting that particularly the ventral synapses are affected. Despite these differences, the density of GlyRs per synapse remained similar.Major comments:1. Line 113: "labeling the β-subunit has proven difficult". This statement is unclear and it would be informative for readers to grasp what exactly has been difficult, and why the approach described here overcomes that? Related to that, the authors state "KI animals reach adulthood and display no overt phenotype, suggesting that the presence of the N-terminal fluorophore does not affect receptor expression and function". That is indeed reassuring, but it does not exclude that receptor numbers, function and distribution are altered. As it seems there is no prior literature on tagging the β subunit, additional evidence that the tag does not interfere with receptor trafficking or functioning would be desirable.

We have clarified why it has been difficult to label the GlyR β subunit until now, lines 113-115 “To date, labeling of GlyRβ in situ using immunocytochemistry has proven difficult due to a lack of reliable antibodies that recognize the native β-subunit (only antibodies for Western blotting recognizing the denatured protein are available), which has severely limited the study of the receptor.” Hence it was important to us to generate this knock-in mouse in order to study the endogenous GlyR at synapses, which is the least well studied receptor mediating fast synaptic transmission.

The reviewer makes an important point regarding the labeling of the GlyRβ-subunit with a fluorescent protein that has also been raised by the other reviewers. We have now verified receptor function by patch clamp recordings of glycine currents in whole-cell configuration in spinal cord neuron cultures from the mEos4b KI mouse (new Supplementary Figure S2C). At saturating glycine concentrations of 300 μM we found no difference in chloride influx between mEos4 KI and WT mice. Since glycine concentrations in the synaptic cleft are in the millimolar range during synaptic transmission, these data strongly suggest that glycinergic transmission is not affected by the presence of the mEos4b under physiological conditions, despite a minor shift in the EC50.

There are several other strong arguments that suggest that mEos4b-GlyRb expression, subcellular localization and function are the same as those of the native subunit. Firstly, the mEos4b sequence was inserted after the signal peptide and before the beginning of the coding sequence of the mature β-subunit (Figure S1). Since the mEos4b sequence does not interrupt the coding sequence it is less likely to affect the receptor conformation. Secondly, we did not notice any behavioural phenotypes in animals carrying the Glrb^Eos^ allele. At the time of weaning, the genotypes of the pups corresponded to the expected Mendelian frequency (new Figure S2A). Moreover, we did not observe a reduction in live expectancy of Glrb^Eos/Eos^ animals (new Figure S2B), demonstrating that the mEos4b-GlyRb does not cause pathology in older animals.

Most importantly, our imaging data (Figure 1-3) provide exhaustive evidence that mEos4b-GlyRb assembles with GlyR α subunits as heteropentameric receptor complexes that are trafficked to the plasma membrane and inserted into the synaptic membrane due to their interaction with the gephyrin scaffold at functional synapses. Using quantitative imaging, we have also shown that homozygous Glrb^Eos/Eos^ KI mice have exactly twice the number of receptors at synapses as heterozygous animals, strongly suggesting no interference in receptor trafficking to the plasma membrane and gephyrin binding. As the mEos4b mice were also bred with the oscillator mouse model of hyperekplexia, which is lethal when homozygous, we could further test the combined effect of Glrb^Eos^ and GlyRa1^spt-ot^. The presence of both alleles did not lead to any noticeable phenotypes in heterozygous oscillator mice. On the contrary, both synaptic targeting and the packing density of the receptors were not altered in this model, despite a region-specific reduction in synapse size due to the reduced availability of the intact GlyRa1 subunit.

We believe that these data overwhelmingly support our conclusion that the presence of the mEos4b tag does not alter the structure and function of the receptor, making this mouse model uniquely suited to study the dynamics and regulation of glycinergic synapses in a quantitative manner and at the molecular level.

2. In the Discussion the authors conclude that “Our quantitative SR-CLEM data lend support to the first model, whereby inhibitory PSDs in the spinal cord are composed of sub-domains that shape the distribution of the GlyRs”. This conclusion seems however based on one example image in Figure 3G that is not very convincing. The EM image seems to show two clearly separated PSDs opposed by two distinct active zones. So, although this conclusion is of high interest, more support should be given to substantiate this conclusion. More general, these subsynaptic domains (SSDs) are hardly further explored, but seem relevant for transmission, particularly given that the synaptic pool of GlyRs at these synapses is not saturated by single release events. How general are these SSDs at these synapses?

The representative image in Figure 3G shows two SSDs within the same postsynaptic site with a continuous presynaptic active zone. It should be noted that the PALM/SRRF images were taken of the entire 2 µm thick slice, whereas the electron micrograph shows only a single 70 nm section. We verified throughout the full 3D stack of serial sections that the presynaptic site remains continuous, which it does. We would also like to point out the scale of the image showing that the two SSDs are only around 170 nm apart, i.e. spatially very close. Our conclusions are however not based on this single image but the whole dataset. The graph in Figure 3I shows 3 synapses (out of N = 36), in which the GlyR density at separate SSDs could be quantified, demonstrating that the receptor density is not different between SSDs. The reviewer is correct that we do not further analyse the SSDs beyond their density and the analysis of the segmentation of the postsynaptic sites (Figure 3E-G). Further work on the functional role of SSDs in synaptic transmission is outside the scope of this manuscript and would indeed merit future study.

3. The approach for counting molecules based on the PALM acquisition has been developed in prior publications and seems robust. It would however be worth to present the reader with a bit more background and explain the assumptions of this approach in more detail. Particularly, since counting of mEos4b can be problematic, as there are multiple dark and fluorescent states of this fluorophore that could be influenced by the illumination scheme, see for instance De Zitter et al., Nat Methods 2019. Since the preceding SRRF acquisition already exposes the fluorophore to high and continuous 561-nm laser power this could skew the counting due to unaccounted conversion and perhaps bleaching of mEos4b. In line with this, although throughout the manuscript the term 'absolute copy numbers' is used the reported numbers are at best an estimate based on a number of assumptions. I think the wording 'absolute numbers' is therefore deceiving and should be nuanced.

We have clarified how the molecule conversion is calculated (Figure S7 legend), to provide a more complete description of the way in which the values were obtained. Further we have explained how we calculated the probability of detection. Since the probability of detection accounts for any unconverted or non-functional mEos4b molecules, our molecule counting approach is relatively resistant to potential pre-bleaching of fluorophores. It should be noted, that 561 nm illumination had no obvious effect on the non-converted (green) mEos4b fluorophores, as judged by the fact that the intensity of receptor puncta was unaffected by the SRRF recordings. We appreciate the reviewers point regarding the term ‘absolute copy number’ and we have adjusted our wording throughout the manuscript accordingly.

4. Related, most of the quantifications are in estimating the number of receptors, and not so much the distribution with the PSD. The term "molecular arrangement" – also used in the title – might therefore be misleading, there is in fact little characterization of how GlyRs are placed within the PSD. More focused analysis quantifying the distribution of receptors within the PSD and/or SSDs would strengthen the manuscript.

By estimating the number of receptors and the exact size of synapses, the main conclusion of our study is that receptor density at dorsal and ventral synapses is identical, independent of synapse size, subdomains, or in fact loss of GlyRs in a mouse model of hyperekplexia. This observation clearly relates to how receptors are packed within synapses, and thus describes their molecular arrangement.

5. The reported N is confusing and makes it hard to judge the reproducibility of the data. Sometimes it refers to number of images, sometimes number of synapses, but it is unclear from how many experiments these are drawn. This should be reported more completely (number of animals should be reported at least) and consistently. In figure 1, the N numbers (N=3-5 images) are particularly low and question how consistent these findings are across multiple animals.

We have clarified the N in the figure legends, to reflect the full size of the datasets that have been analysed.

6. The levels of mRFP-Gephyrin seem to differ between the different mouse lines, is this a significant difference?

No significant differences in mRFP-gephyrin levels were found in animals with different mEos4b-GlyRb genotype (Figure 1B). However, expression of mRFP-gephyrin in heterozygous animals is 50% of that in homozygous mRFP-gephyrin KI animals (not shown).

7. The ICQ analysis for co-localization is hardly explained. How do we interpret this parameter? What does an average value of ~0.3 mean? A comparison with sets of proteins that do not overlap as a negative control would strengthen the conclusion.

We have clarified that an ICQ value of 0.3 is indicative of a very high spatial correlation between pixels, and provided a corresponding reference for ICQ analysis (lines 209-210). We would like to point out that the scale of the ICQ is between -0.5 to 0.5, meaning that a value of 0.3 comes close to complete correlation.

Minor comments:1. "Very little fluorescence was detected in the forebrain, despite the high reported expression of the Glrb transcript". Can the authors expand on this? What would explain this discrepancy?

We have clarified the text to include “suggesting that protein levels are controlled by post-translational mechanisms in a region-specific manner, as previously proposed (Weltzien et al., 2012)” (Lines 152-153). The reason for this discrepancy is not known. However, the distribution of mEos4b expression throughout the brain is as expected, based on the literature.

2. "What region is quantified in Figure 1B? is the same region in all conditions? This should be specified more clearly as the manuscripts presents a clear gradient in expression levels in the spinal cord and thus the location will influence the intensity measurements.

We have explained in the text that this is the region at the centre of the ventral horn identified by the white square in Figure 1A, and that the same region was analysed for all images across all animals. Page 5, lines 160-161 “The same region of the ventral horn, indicated by the white square in Figure 1A was taken for quantification of mEos4b-GlyRβ and mRFP-gephyrin expression in all conditions.”

3. The labeling approach does not differentiate between surface and internal receptors, this should be made more explicit in the text.

Whilst this is correct, we have only analysed mEos4b-positive synapses that had corresponding gephyrin clusters, meaning synapses where receptors are located in the postsynaptic membrane. Indeed we found that all mEos4b clusters imaged colocalised with mRFP-gephyrin clusters. We have adjusted the text accordingly, page 6, line 205-206 “All mEos4b-GlyR clusters closely matched the mRFP-gephyrin clusters, confirming the localization of the receptors in the postsynaptic membrane.”

Significance:The presented data are interesting and the experiments are technically advanced and carefully performed. Particularly the SR-CLEM approach is technically advanced. The datasets present a quantitatively detailed characterization of spinal cord synapses and will be of interest for researchers working in the field of spinal cord circuitry, as well as super-resolution imaging. The conceptual advance for the field is however somewhat limited. It seems that the presented data confirm the general notion that receptor numbers and synapse size are highly correlated. So, although this manuscript describes very interesting observations, in its present form the manuscript does not provide any new mechanistic insight or significant advance in our understanding of how these synapses operate.

We thank the reviewer for his/her comments relating to the technicality of our manuscript. However we think that the statement “The conceptual advance for the field is however somewhat limited” is unfair, as this level of organisation of inhibitory synapses at the molecular scale has never been achieved before, as pointed out by the other reviewers, and especially not as regards different ages of animals and a disease model that directly affects receptor numbers in a region-specific manner. We therefore believe that our study will have a substantial impact within the fields of synaptic neuroscience as well as quantitative neurobiology.

Referee cross-commenting:I agree with the other reviewers that this study is technically advanced, but I remain critical towards the extent of conceptual advancement this study brings and there are some important concerns with the presented data that need to be addressed. Nevertheless, indeed many of these concerns can be addressed without additional experiments. As pointed out also by other reviewers additional validation that the fusion proteins are not disrupting their function or organization would be important.Reviewer 2:Maynard et al. investigate (inhibitory) glycinergic synapses in mouse spinal cord, which regulate motor and sensory processes. The authors analyse the molecular architecture and ultra-structure of these synapses in native spinal cord tissue using quantitative super-resolution correlative light and electron microscopy. The major finding is that GlyRs exhibit equal receptor-scaffold occupancy and constant absolute packing densities across the spinal cord and throughout adulthood, although ventral and dorsal inhibitory synapses differ in size. Moreover, what the authors call a „stereotypic arrangement" is even maintained in a hypomorphic mutant (oscillator), which is deficient in the adult GlyR a1 subunit.Specific comments:1. To reach their conclusions the authors generate two knock-in mouse lines, one with mEOS-labelled GlyR ß-subunit and one with mRFP-labelled gephyrin, a subsynaptic scaffolding protein of inhibitory synapses, which are subsequently crossed. Both changes are not unproblematic, as mutations in the N-terminal end of the GlyR ß subunit polypeptide chain might interfere with the assembly of functional GlyR (consisting of a und ß subunits) and mutations at the N-terminal end of gephyrin interfere with its homo-oligomerization into higher molecular assemblies.

We have demonstrated that the function of mEos4b-GlyRb does not differ significantly from WT GlyRs, by carrying out electrophysiological experiments (new Figure S2C). For a detailed response, please see the response to the first comment of reviewer 1. The mRFP-gephyrin KI strain has been validated and published previously (see Machado et al., 2011, J Neurosci; Specht et al. 2013 , Neuron) and was not specifically generated for this study. The experiments with the oscillator mutant did not include the mRFP-gephyrin allele. In these experiments, the wildtype Glrb^Eos/Eos^ (Figure 4, 5) behaves exactly as the Glrb^Eos/Eos^ in the double knock-in (Figure 1, 2), further validating the mouse models used.

2. However, in this experimental design both labelled proteins reach postsynaptic membrane specialisations. In case of the ß-subunit quantitative evaluation confirms that heterozygous animals contain only half of the labelled protein as homozygous, which is an indication but not a proof that the correct stoichometry of adult GlyR is maintained. Likewise, mRFP-labelled gephyrin assembles with WT-gephyrin in subsynaptic domains, but it is not clear, if the size and density of the synapses is changed by the knock-in procedure as compared to WT-synapses.

An effect of the mRFP tag on gephyrin clustering can be ruled out, since we observed no difference in synapse size and receptor density in Glrb^Eos/Eos^ animals with (Figure 1, 2) and without the Gphn^mRFP^ allele (Figure 4, 5, oscillator wild-type controls). Similarly, the synaptic mEos4b-GlyRb levels in heterozygous animals were precisely half those of the homozygous animals, strongly suggesting that the expression and trafficking of the tagged receptor subunit is unchanged, as the reviewer acknowledges. In the absence of any obvious behavioural and/or functional phenotypes (Figure S2) this KI model is in our view is an exceptional tool to study GlyRs expressed at endogenous levels in a cell-type specific manner.

3. Accepting these constraints, which to the knowledge of this reviewer have never been addressed to satisfaction, the authors provide a technically excellent, comprehensive analysis of glycinergic synapses in the spinal cord of double knock-in mice. Therefore, it should be stated in the title, that the investigations were performed with double knock-in instead of „native" spinal cord. Text and figures are clear and accurate and represent the state of the art.

We thank the reviewer for the positive comments regarding the techniques used in the study, and the clarity of the text and figures. We have adjusted the title as requested.

4. Finally, the reviewer would like to raise a minor point: the term postsynaptic density is derived from electron microscopical studies of synapses, where asymmetrical synapses display a „postsynaptic density" but symmetrical synapses do not. The latter were identified as inhibitory synapses and therefore, by definition, inhibitory synapses do not have a postsynaptic density, but rather a postsynaptic membrane specialisation. The use of the term „postsynaptic density" should, therefore, be restricted to excitatory synapses.

We are conscious of the importance of correct definitions and have revised the terminology, referring to “postsynaptic sites”, “postsynaptic domains”, and “postsynaptic specializations” as appropriate throughout the manuscript.

Significance:The authors provide a state of the art advanced light and electron microscopical analysis of glycinergic synapses in the mouse spinal cord. They suggest a robust “stereotypical” mechanism in place, which guarantees a fixed stoichiometry of relevant components, which is even maintained in a hypomorphic mutant, which is believed to represent a mouse model of human hyperekplexia (startle disease).Referee cross-commenting:I would like to corroborate the arguments of the previous reviewer: it is not clear to which extent the fusion proteins influence the measurements, which are technically very advanced and well done, however. The authors do definitely not investigate "native spinal cord" as stated in the title.The argument concerning fusion proteins must be taken especially serious as the fusions were induced in regions known to be responsible for assembly of glycine receptors and oligomerization of gephyrin.

We have verified the receptor function with electrophysiological recordings and clarified exactly where the fluorescent protein was inserted (see reviewer 1 response). Given the similarity in synapse size, fluorescence intensities and molecule densities observed in neurons expressing different combinations of tagged and native receptors and scaffold proteins, we strongly believe that all animal models used are well suited to the experimental aims of our study.

Reviewer 3:Glycinergic synapses are the least well understood of synapses that mediate fast synaptic transmission. The manuscript by Maynard et al. adds new information about the structural aspects of these synapses, using PALM and EM imaging of spinal cord synapses from mice at 2 and 10 months. The authors created a knock-in mouse that expresses a tagged GlyRβ subunit, allowing synaptic localization of glycine receptors; all synaptically localized glycine receptors are thought to require the β subunit to be tethered by gephyrin. The authors compare synaptic profiles from: 2 month old vs. 10 month old mice; dorsal vs. ventral horn; and GlyR1-reduced vs. wild type mice. Strikingly, they find a tight relationship across all of these variables between glycine receptor puncta and gephyrin puncta, as well as an apparently constant "packing density" of glycine receptors. They conclude that synaptic extent is likely to be the most important determinant of synaptic strength, as the density of receptors within the postsynaptic density is constant. These results use cutting-edge imaging and are analyzed with care, and add new information to our understanding of these relatively less well characterized synapses.Major comments:1. The key conclusions are convincing and the claims appear solid. Additional experiments are not needed to support these claims. The data and the methods are largely presented in such a way that they can be reproduced, although there are minor suggestions for improvement below.

We thank the reviewer for his/her positive comments.

Minor comments:1. Do the authors have any comment on the requirement during, e.g. LTP, for insertion of a gephyrin-GlyR unit? The lead author has speculated that gephyrin creates "slots" for GlyRs; yet apparently each slot is already filled in the snapshots taken here. How might postsynaptic LTP occur (Kandler group, Kauer group papers)?

Given the reciprocity of GlyR and gephyrin clustering at synapses, the occupancy of binding sites (and in turn the number of available ‘slots’) is dependent on the strength of receptor-scaffold interactions, as discussed previously (Specht 2020, Neuropharmacol). In this study we demonstrate that the density of GlyRs at synapses is constant, which implies that the receptor occupancy is also the same, with the possible exception of mixed inhibitory synapses in the superficial dorsal horn that contain a majority of GABA_A_Rs. The PALM/SRRF data are represented as rendered image reconstructions and not as pointillist representations, and the detection of unoccupied binding sites is below the spatial resolution of our approach. However, the high spatial correlation of the signal intensities (ICQ ≈ 0.3) suggests that receptor occupancy is equal between and within synapses. It has previously been established that there are more scaffold proteins than receptors at synapses (Specht et al. 2013, Neuron; Patrizio et al. 2017, Sci Rep). Based on these studies we report that approximately half the gephyrin binding sites are occupied by receptors (lines 262-655). We have also expanded the discussion, describing how shape and size of synapses may affect synaptic transmission, as well as the possible role of receptor-gephyrin interactions in synaptic plasticity at glycinergic synapses.

2. It would be very interesting in the discussion to contrast the present observations with what is known about excitatory synapses (NMDA and AMPAR distributions) and GABAergic synapses. Are the authors at all surprised that receptor packing is constant across conditions? Can the authors speculate on how non-gephyrin binding receptors (homomeric α receptors, which are found in recordings) may function and be tethered to the membrane.

We have included additional information about receptor numbers and distributions at excitatory (lines 428-438) and GABAergic (lines 389-393) synapses in the discussion. So far, homomeric GlyRs composed of α subunits have been found to be exclusively extrasynaptic. As stated on page 4, lines 111-112 the β subunit is required for binding of the GlyR to gephyrin and subsequent anchoring at the synapse. Previous studies have shown exocytosis of receptors to occur at extrasynaptic sites followed by lateral diffusion to synapses. Homomeric GlyRs are therefore most likely targeted to the extrasynaptic plasma membrane where they remain due to the lack of the β subunit.

3. Figure S1. It would be most helpful to quantify this; at the least to include an atlas-like drawing to allow identification of the structures illustrated and containing Glrb; better yet would be quantification of staining in regions where this is strongest.

We have added an atlas indicating the different brain regions expressing mEos4b-GlyRb protein as a new Supplementary Figure S3. The regional expression pattern agrees with the available literature about protein expression of the GlyRb subunit in different brain regions and hence provides further evidence that mEos4b-GlyRb is expressed like the native receptor. Due to the relatively low resolution of the tiled image no accurate quantification was possible. We have however added higher magnification confocal images of representative brain regions expressing varying amounts of GlyRb.

4. The fact that the lower panel in B is labeled as +/+ across all groups is initially confusing; perhaps relabel as mEos4 -/-, +/- and +/+?

We assume that the reviewer is referring to Figure 1B. The genotype of both the Glrb^Eos^ and the Gphn^mRFP^ allele is now indicated on the x-axes, and the legend has been modified to clarify that all these animals were homozygous for Gphn^mRFP/mRFP^. We have strived to remain consistent throughout the manuscript when referring to genotypes and protein levels.

5. Do gephyrin levels drop in WT mice as well as in the mEosr-GlyRb mouse between 2 and 10 months? Do the authors have any thoughts on this (Supp figure S2)?

We found no differences in gephyrin levels between 2 and 10 months. Figure S2 (now Figure S4C) shows the number of synaptic gephyrin clusters, which was the same at different ages and genotypes.

Significance:Glycinergic synapses are the least well understood of synapses that mediate fast synaptic transmission. The manuscript by Maynard et al. adds new information about the structural aspects of these synapses, using PALM and EM imaging of spinal cord synapses from mice at 2 and 10 months. The authors created a knock-in mouse that expresses a tagged GlyRβ subunit, allowing synaptic localization of glycine receptors.This will be of interest to those studying inhibitory synapses, and more broadly to synaptic morphologists, physiologists and imagers for comparison with other synapse types.My own expertise is NOT in these techniques, but I am a synaptic physiologist with a standing interest in glycinergic synapses; thus I am not providing serious technical critiques.Referee cross-commenting:Hi all, I agree with the other two reviewers, and do not have anything else to add.Reviewer 4:The authors used a correlative approach and combined photo-activated localization microscopy with electron microscopy to characterise Glycinergic synapses in spinal cord tissue. Some of the major findings are:– The receptor-scaffold occupancy and packing densities of glycinergic synapses in different regions of the spinal cord are the same.– Gephyrin clusters in the spinal cord are composed of sub-domains that shape the GlyR clusters.– Ventral horn synapses are generally larger, more complex (containing a number of gaps) and contain more GlyRs.–In a mouse model of Hyperekplexia, the number of GlyRs is reduced resulting in smaller synapses in the ventral spinal cord.Major comments:1. Are the key conclusions convincing?

Yes.

2. Should the authors qualify some of their claims as preliminary or speculative, or remove them altogether?

No.

3. Would additional experiments be essential to support the claims of the paper? Request additional experiments only where necessary for the paper as it is, and do not ask authors to open new lines of experimentation.

No.

4. Are the suggested experiments realistic in terms of time and resources? It would help if you could add an estimated cost and time investment for substantial experiments.

N/A.

5. Are the data and the methods presented in such a way that they can be reproduced?

Yes.

6. Are the experiments adequately replicated and statistical analysis adequate?

Yes.

Minor comments:1. Specific experimental issues that are easily addressable.

Please see below.

2. Are prior studies referenced appropriately?

Yes.

3. Are the text and figures clear and accurate?

Yes.

4. Do you have suggestions that would help the authors improve the presentation of their data and conclusions?

Please see below.

5. As the authors pointed out, fusing mEos to the extrasynaptic terminal of GlyRb has been difficult and therefore this construct would benefit the larger scientific community. Figure 1C is a nice imaging control for expression efficiency, however, it is in stark contrast with the lack of functional control. Do authors have any electrophysiological evidence showing that the insertion of mEos4b doesn't modulate channel function? I would assume that the construct would be tested in cell lines before the KI mouse line was created. Was any functional analysis done? If yes, it would be very useful to show it. I do appreciate that the authors used a standard insertion between the 4th and 5th AA in the extracellular domain, which in most cases does not abolish channel function. Given the lack of an obvious phenotype in the KI mouse model, I believe that this is also the case here. However, I disagree with the statement in lines 120-121: "the presence of the N-terminal fluorophore does not affect receptor expression and function." I believe that if there are no electrophysiological measurements of GlyR function, this statement remains speculative. As the authors pointed out in their previous publication: "receptor function and gephyrin binding are not independent properties. Instead, we think that conformational changes triggered at extracellular or intracellular protein domains have downstream consequences on channel opening as well as receptor clustering." In line with this, my concern is that the modulation of channel function by mEos4b could result in an altered cluster size at synapses. There is a large body of literature showing that just one missense mutation in the extracellular domain of ion channel subunits can lead to synaptopathies because the channel function gets modulated, and there is an abundance of similar examples involving mutations of GlyR and GABAAR subunits. In my view, comparing the function of GlyRs incorporating wt-GlyRb and mEos4b-GlyRb subunits is important for the correct interpretation of the main findings of this work and would strengthen the publications.

As the reviewer points out, the insertion of the mEos4b sequence was considered carefully in order to have the least impact on receptor function. GlyR channelopathies are often caused by point mutations within the coding sequence, which is not the case in the Glrb^Eos^ allele. Instead, the mEos4b sequence was inserted after the single peptide of GlyRb, duplicating several amino acid residues in order to maintain the correct cleavage site and N-terminus of the mature receptor, and to not interrupt the GlyRb coding sequence (Figure S1B). In order to verify that the mEos4b-tag does not affect GlyR function, we have now carried out electrophysiological experiments (new Figure 2C). For a detailed description please see the response to the first comment of reviewer 1.

6. Line 189: Are the authors making conclusions based on intensity comparison of red mEos4b and mRFP? The title of this section implies that the red form of mEos was compared to mRFP(?) But mEos converts from green to red only partially. Was the probability for conversion taken into account at this point? Please clarify which version of mEos was compared to mRFP.

In line 189 (now 218) we compared the intensities of mRFP-gephyrin with those of converted (red) mEos4b in SRRF / PALM super-resolution images of the synapses (Figure 2D). Since the absolute intensities are altered by the process of image reconstruction, the probability that mEos4b is photoconverted does not have to be taken into account. The constant ratio of the SRRF and PALM image intensities confirms the data in Figure 1D showing that GlyR and gephyrin amounts are highly correlated throughout the spinal cord (with the exception of the superficial layers of the dorsal horn). We have clarified in the text that this analysis was carried out on reconstructed SRRF images of mRFP-gephyrin and PALM images of mEos4, line 202.

7. Line 192: Please clarify how the density threshold was calculated/determined? This is important for the replication of the experiments, and it also has implications for the calculated probability of detection of mEos4b. I am not aware that this probability was calculated before for mEos4b and therefore other researchers may decide to rely on the value calculated here.

We have now clarified in more detail how the probability of detection was calculated (new Supplementary Figure S7 legend).

8. In Figure 2 Gephyrin clusters look consistently smaller than GlyR clusters, which is inconsistent with the published work. I assume that the difference in size is a consequence of different image reconstruction methods(?) However, I would assume that SRRF would have lower resolution than your PALM measurements and that would result in wider Gephyrin clusters. Could you please explain this discrepancy? Also, could you provide an estimate for the image resolution in SRRF and PALM techniques? For SMLM, localization precision would suffice.

We have provided an estimate of the resolution of the two techniques using Fourier ring correlation, which gave 46 nm for SRRF and 21 nm for PALM. Additionally we have precised the discrepancy between reconstruction methods, page 6, lines 194-200 “The spatial resolution was estimated using Fourier ring correlation (FRC), which measures the similarity of two images as a function of spatial frequency by comparing the odd and even frames of the raw image sequence. According to this analysis, the spatial resolution of SRRF was 46 nm and that of PALM 21 nm. It should be noted that the synaptic puncta in the SRRF images appear somewhat smaller and brighter due to differences in the reconstruction methods that result in differences in the dynamic intensity range.”

9. Why is the data in Figure 5D and E represented as Detections/Synapse instead of GlyRs/Synapse? Could you please re-plot this so that a comparison with Figure 2H and I is straightforward?

We have converted the detections to receptor copy numbers as requested (Figure 5D,E).

10. Figure S5C: for P=0.5, 2=0.25. Please correct. Also, I assume that the second graph is what would be observed experimentally for dimers and P=0.5. Please clarify in the figure caption.

This was a mistake and has been corrected. We have also clarified which parts of the calculations are theoretical and which values were derived from our experimental data. We have provided a more detailed description in the figure legend of Supplementary Figure S7.

11. Line 606: Please provide a complete derivation of this formula.

We have provided a full derivation of this formula (new Figure S7C).

Significance:The work described here seem to be a natural progression of a publication by Patrizio et al., 2017 that came out from the same laboratory. This study uses advanced methodologies in the imaging space to visualise and characterise Glycinergic synapses in spinal cord tissue. The experiments described here are technically demanding as evidenced by the relatively small number of publications describing super-resolution measurements in tissue samples. Even more rare are studies that attempt to do single protein counting in neuronal culture and tissue sections. Therefore, I believe that this work brings significant technical advancement in the field of super-resolution and corelative microscopy. The findings are also highly significant for all fields of neuroscience in which the structure of inhibitory Glycinergic synapse is relevant, ranging from the fundamental understanding of inhibitory synapse function to pathologies involving Glycinergic signalling.I have substantial experience in different microscopy methods, including quantitative super-resolution microscopy based on single molecule counting. My background also covers the structure and function of GABAA and Glycine receptors using electrophysiology. I am familiar with the methods used in electron microscopy and the process of creating KI mouse lines, however I don't have hands-on experience in these fields.